**Subject Category:**
Biology (whole organism)

ecology/behaviour

winter migration, trophic ecology, *Balaenoptera musculus*, *Balaenoptera physalus*, *Balaenoptera borealis*, stable isotopes

**Author for correspondence:**
Mónica A. Silva
e-mail: monica.silva.imar@gmail.com

# Stable isotopes reveal winter feeding in different habitats in blue, fin and sei whales migrating through the Azores

Mónica A. Silva[1,2,3], Asunción Borrell[4], Rui Prieto[1,3], Pauline Gauffier[4], Martine Bérubé[5,6], Per J. Palsbøl[5,6] and Ana Colaço[1,3]

[1]Okeanos Centre & Institute of Marine Research (IMAR), University of the Azores, 9901-862 Horta, Portugal
[2]Biology Department, Woods Hole Oceanographic Institution, Woods Hole, MA 02543, USA
[3]Marine and Environmental Sciences Centre Açores (MARE), Department of Oceanography and Fisheries, 9901-862 Horta, Portugal
[4]Institute of Biodiversity Research (IRBio) & Department of Evolutionary Biology, Ecology and Environmental Sciences, Faculty of Biology, University of Barcelona, 08028 Barcelona, Spain
[5]Groningen Institute of Evolutionary Life Sciences, University of Groningen, 9747 AG Groningen, The Netherlands
[6]Centre for Coastal Studies, Provincetown, MA 02657, USA

MAS, 0000-0002-2683-309X

Knowing the migratory movements and behaviour of baleen whales is fundamental to understanding their ecology. We compared $\delta^{15}N$ and $\delta^{13}C$ values in the skin of blue (*Balaenoptera musculus*), fin (*Balaenoptera physalus*) and sei (*Balaenoptera borealis*) whales sighted in the Azores in spring with the values of potential prey from different regions within the North Atlantic using Bayesian mixing models to investigate their trophic ecology and migration patterns. Fin whale $\delta^{15}N$ values were higher than those recorded in blue and sei whales, reflecting feeding at higher trophic levels. Whales' skin $\delta^{15}N$ and $\delta^{13}C$ values did not reflect prey from high-latitude summer foraging grounds; instead mixing models identified tropical or subtropical regions as the most likely feeding areas for all species during winter and spring. Yet, differences in $\delta^{13}C$ values among whale species suggest use of different regions within this range. Blue and sei whales primarily used resources from the Northwest African upwelling and pelagic tropical/subtropical regions, while fin whales fed off Iberia. However, determining feeding habitats from stable isotope values remains difficult. In conclusion, winter feeding appears common among North Atlantic blue, fin and sei whales, and may play a crucial role in determining their winter distribution. A better understanding of winter feeding behaviour is therefore fundamental for the effective conservation of these species.

# 1. Introduction

Migration is generally defined as a regular, long-distance pattern of movement between spatially disjunct habitats. It is a common population-level strategy across many taxa that emerges from individuals' movements that are probably aimed at maximizing fitness [1]. The individuals' migration behaviour, timing, routes and destination can affect their survival, foraging and breeding success, physiological condition, as well as exposure to predators and anthropogenic threats [2–5]. Consequently, identification of migration patterns is fundamental to understanding population dynamics and susceptibility to anthropogenic threats and global environmental change [6].

The classical model of baleen whale migration assumes that most individuals migrate between high-latitude productive summer feeding areas and subtropical to tropical breeding winter areas, where feeding is believed to be absent or severely reduced [7–9]. However, as the capacity to observe and follow whale movements and behaviour increases, so does the notion that migration patterns vary among species, populations and individuals. In some populations of humpback (*Megaptera novaeangliae*), fin (*Balaenoptera physalus*) and blue (*Balaenoptera musculus*) whales, some individuals remain at high to mid-latitudes during the breeding season, or in tropical waters during the feeding period [10–14]. Other populations may be completely non-migratory, such as the Mediterranean and Gulf of California fin whales, Sri Lankan blue whales and Arabian Sea humpback whales [15–18]. Similarly, there have been a number of reports of baleen whales foraging during migration or on the breeding range [19–21]. However, for most baleen whale populations, the migration destinations and seasonal feeding behaviour remain unknown.

Earlier studies [19,22] employed satellite tags to examine the movements and behaviour of North Atlantic blue, fin and sei (*Balaenoptera borealis*) whales encountered off the Azores. Blue and fin whales suspended their northward migration and remained at temperate latitudes for periods of up to a few months. While at mid-latitudes, individuals spent a substantial proportion of their time engaged in behaviours inferred as foraging [19]. In contrast, sei whales left the Azores immediately after tagging and continued their northbound migration towards the Labrador Sea [22]. The movements obtained from the satellite data agreed well with photo-identification and behavioural data collected in the Azores indicating that sei whales have short residency times near the islands and rarely engage in foraging (M.A.S. & R.P. 2015, unpublished data). These results suggest different migration strategies in sei whales relative to blue and fin whales with respect to seasonal feeding–fasting cycles. However, this observation was based upon a brief snapshot during the spring migration and from a small number of individuals.

Indeed, despite substantial progress in tracking technology, deploying satellite tags in baleen whales that remain on the animal for the entire annual migratory cycle remains challenging [23,24].

Stable isotope analysis (SIA) can partially circumvent these challenges and yield insights into the migration and trophic ecology of whales. Such an approach is based upon two premises: the stable isotope composition of an organism reflects that of its food [25], and the isotopic composition of marine primary producers varies depending on different oceanographic and biogeochemical processes, producing unique geographical distributions, so-called isoscapes [26,27].

The ratios of heavy to light isotopes of elements in the tissues of consumers reflect those of the food assimilated over a period of time (from days to years, depending on tissue-specific turnover rates), with a slight difference [28]. This difference, termed discrimination factor, results from the selectivity for the lighter isotopes during consumers' metabolic processes [29,30]. Although discrimination factors are tissue and taxa specific, and may be influenced by an array of environmental, biological and physiological factors [28], differences between consumer tissues and diet usually vary in a predictable manner across the food chain. Nitrogen isotope ratios ($^{15}N/^{14}N$, $\delta^{15}N$) usually increase by 2–4‰ per trophic level [30,31]; hence the $\delta^{15}N$ value is commonly used to infer the trophic positions of consumers [32]. In contrast, carbon isotope ratios ($^{13}C/^{12}C$, $\delta^{13}C$) change little (approx. 1‰) with trophic level but can be a useful indicator of the spatial origin of carbon assimilated by consumers [31,33].

$\delta^{13}C$ values of primary producers are strongly influenced by the $\delta^{13}C$ value of dissolved inorganic carbon, temperature, $CO_2$ drawdown, as well as the size, composition and growth rates of the phytoplankton community [27,34]. Consequently, phytoplankton $\delta^{13}C$ values vary substantially with latitude, and among primary production pathways, such as phytoplankton (−24‰ to −18‰), macrophytes (−27‰ to −8‰) and seagrasses (−15‰ to −3‰), and productivity regimes (upwelling phytoplankton: −18‰ to −16‰; open ocean phytoplankton: −22‰ to −18‰) [27,31,35,36]. In general, $\delta^{13}C$ values tend to decline from coastal and benthic sources to offshore and pelagic sources, and from northern to southern latitudes within each hemisphere [27]. $\delta^{15}N$ also varies geographically, depending on the sources of nitrogen used by primary producers, nitrogen cycling and assimilation processes [37–39]. In the Atlantic Ocean, $\delta^{15}N$

values of zooplankton in the upper 200 m decrease with latitude and reach the lowest values in oligotrophic subtropical and tropical waters [27].

In baleen whales, analyses of $\delta^{15}N$ and $\delta^{13}C$ values in biopsy-derived skin samples and faecal material have been employed to infer diet and trophic level, niche partitioning within and among species, as well as population structure and migratory movements (e.g. [21,40–45]). With the development of Bayesian isotope mixing models, the potential of SIA to investigate the spatial and trophic ecology of marine predators improved substantially. These models enable the contribution of different stable isotope sources to the diet to be quantified, and the associated uncertainty to be estimated [46].

In this study, SIA and Bayesian mixing models were used to assess (i) if blue, fin and sei whales feed at lower latitudes and (ii) their habitat use during winter and spring. For that, we estimated the $\delta^{15}N$ and $\delta^{13}C$ values in skin biopsies collected from blue, fin and sei whales off the Azores during their northward spring migration. Isotopic incorporation rates of baleen whale skin have been poorly studied. The rate at which dietary isotopes are incorporated into consumer tissues depends on the rate of tissue synthesis and replacement [47]. Half-life turnover in the delipidated skin of captive bottlenose dolphins (*Tursiops truncatus*) was estimated at approximately 24 days and approximately 48 days for $\delta^{13}C$ and $\delta^{15}N$ values, respectively, and complete turnover in approximately 100 days for $\delta^{13}C$ and approximately 200 days for $\delta^{15}N$ values [48]. Turnover rates in large baleen whales are expected to be longer than those in bottlenose dolphins because of their large body mass and lower metabolic rate [47,49]. In fact, in blue whales, the mean skin turnover of $\delta^{15}N$ values was estimated at 5.4 months [50] and, although corresponding figures for the $\delta^{13}C$ turnover could not be assessed, by similarity to dolphins it is reasonable to assume that they may be in the range of approximately three months. Therefore, we assumed that skin samples collected mostly between March and June reflected isotopic ratios acquired during early winter to early spring. In addition to the skin samples, we also analysed the $\delta^{15}N$ and $\delta^{13}C$ values in a small number of faecal samples collected opportunistically. Stable isotopes in whale faeces represent a record of recently (from hours to 1–2 days) ingested prey [40] and, in this study, these few samples were used to gain insights into the diet of whales foraging in the offshore waters around the Azores.

# 2. Material and methods

## 2.1. Sample collection

Skin and faecal samples from blue, fin and sei whales were collected between 2002 and 2014 in the Archipelago of the Azores, Portugal (figure 1). Skin samples were collected using a hollow-tipped biopsy dart fired from a crossbow and stored in sterilized Eppendorf tubes. Faecal samples were collected opportunistically using a nylon stocking secured over a metal ring attached to a pole. The faeces were present as chunks of varying sizes and were randomly sub-sampled for analysis. The salt water was drained off the faeces and each sub-sample was stored in a plastic bag. Skin and faecal samples were placed in a cooler on ice-packs until landfall. The date, time, geographical position, behaviour and approximate body length of the sampled whale, as well as group size, were recorded. In order to avoid including duplicate samples from the same individual, whales were photographed using digital cameras for later identification using standard methods [51–53]. Each skin sample was separated into two sub-samples for SIA and molecular sexing. Isotopic (skin and faecal) samples were frozen at –80°C, while skin samples for molecular analysis were preserved in 90% ethanol and stored at 4°C.

## 2.2. Stable isotope analyses

Lipids are depleted in $^{13}C$ and typically have $\delta^{13}C$ values that are more negative than those of proteins and carbohydrates within an organism [29], which can introduce considerable bias in SIAs. A common approach to deal with the potential lipid effect on $\delta^{13}C$ values is to remove lipids from tissues by chemical lipid extraction [32]. However, lipid extraction may also alter the $\delta^{15}N$ values of tissues, although the magnitude and direction of change is species and tissue dependent. In the case of baleen whale skin, some studies reported a significant decline in $\delta^{15}N$ values after lipid extraction [43], while other studies documented a significant increase or no change in $\delta^{15}N$ values for the same whale species [54]. Similarly to what was observed in the muscle tissue of terrestrial mammals [55], changes in $\delta^{15}N$ in baleen whale skin due to lipid extraction were generally small and within the analytical precision typical of $\delta^{15}N$ measurements [43,54]. Nonetheless, a way to avoid potential effects on $\delta^{15}N$ is to measure $\delta^{13}C$ in lipid-extracted tissue and $\delta^{15}N$ in non-extracted tissue [54]. Obviously, this

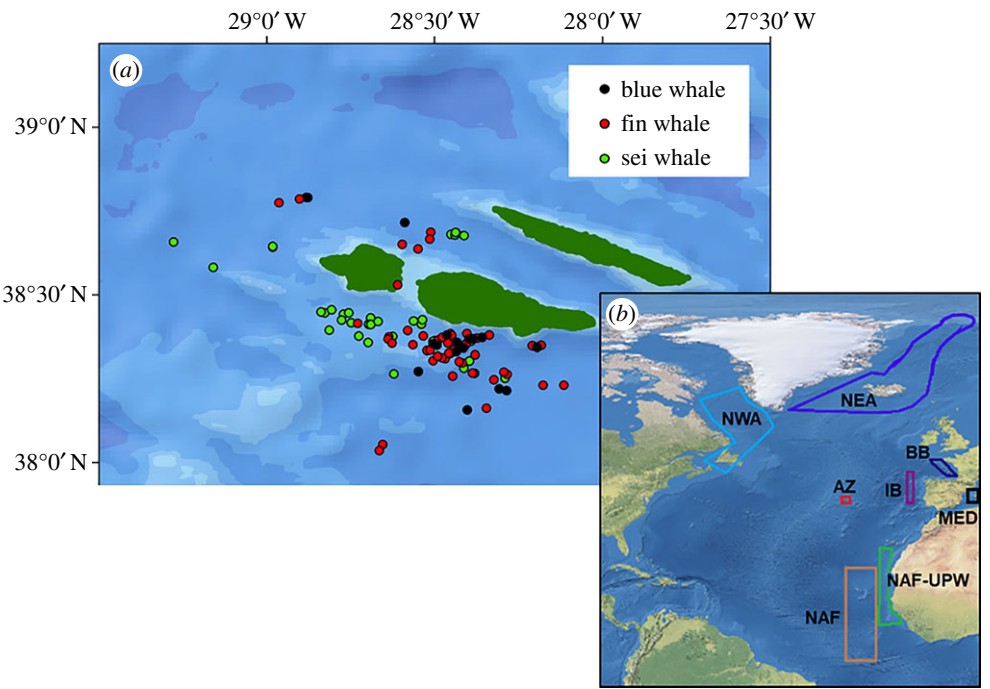

**Figure 1.** Location of (*a*) skin and faecal samples of blue, fin and sei whales collected in the Azores, and (*b*) sampling ranges of potential prey compiled from the literature (electronic supplementary material, table S1) in each region. Regions are: AZ, Azores; IB, Iberia; BB, Bay of Biscay; NAF, North Africa; NAF-UPW, North Africa upwelling; NEA, Northeast Atlantic; NWA, Northwest Atlantic; MED, Mediterranean.

requires that a larger amount of sample is available to conduct separate analyses. Unfortunately, this was not the case in the present study, where each skin sample was already split for SIA and genetic analyses.

Hence, lipids were extracted from skin samples using repeated rinses with 2 : 1 chloroform : methanol solution [43] and $\delta^{15}N$ and $\delta^{13}C$ were analysed in lipid-extracted skin. Lipid-extracted samples were freeze dried and ground with a mortar and pestle into a homogeneous fine powder. Approximately 1 mg of powder sample was used for $\delta^{13}C$ and $\delta^{15}N$ measurements. Analyses were carried out using isotope ratio mass spectrometers (Thermo Finnigan Delta Plus XP IRMS and Finnigan MAT Delta C) coupled to a Costech 4010 or a Flash EA-1112 elemental analyser. Stable isotope ratios were expressed conventionally as $\delta$ (‰) relative to the Pee Dee Belemnite (PDB) standard for the $\delta^{13}C$ value and atmospheric nitrogen (AIR) for the $\delta^{15}N$ value.

## 2.3. Molecular sexing

Genomic DNA was extracted using QIAGEN DNEasy™ extraction columns for animal tissue, following the manufacturer's instructions (QIAGEN Inc.). The quality of the DNA was checked visually by electrophoresis through a 0.7% agarose gel and quantified with a Qubit™ following the manufacturer's instructions (Thermo Fisher Scientific Inc.). DNA extractions were normalized to a concentration of 10 ng DNA $\mu l^{-1}$. The sex of each sampled whale was determined by the amplification of the ZFY and ZFX specific sequences [56].

## 2.4. Statistical analysis

Only isotopic ratios measured from whale skin were used in the following analyses. For each whale species, a permutational multivariate analysis of variance (PERMANOVA) [57] was used to assess the effects of sex, season and year on the combined $\delta^{15}N$ and $\delta^{13}C$ values. The PERMANOVA is a non-parametric method that analyses the variance of multivariate data based on pairwise measures of dissimilarity or distance [57]. The PERMANOVA analyses were performed on a matrix of normalized Euclidean distances of $\delta^{15}N$ and $\delta^{13}C$ values, with three fixed factors (interactions between factors could not be considered because of small sample sizes) and employing type III sum of squares. Only years with sample sizes greater than or equal to 2 were included in the analysis. Subsequently, univariate ANOVAs, followed by Tukey's *post hoc* tests, were conducted to further assess the effects of

significant variables identified in the PERMANOVA analysis on $\delta^{15}N$ and $\delta^{13}C$ values separately. Prior to analyses, the assumptions of data normality and homogeneity of variances were checked using Kolmogorov–Smirnov and Levene's tests, respectively.

Differences in $\delta^{15}N$ and $\delta^{13}C$ values between whale species were assessed using a one-way PERMANOVA with type III sum of squares using the normalized Euclidean distances. Standard ellipse areas corrected for small sample size ($SEA_C$) were estimated to determine the isotopic niche space of each whale species and assess overlap in niche space among them. The $SEA_C$ represents 40% of the total area encompassing all samples in the $\delta^{13}C$–$\delta^{15}N$ biplot space and is less sensitive to outliers and unbiased with respect to sample sizes [58]. The percentage of overlap in $SEA_C$ was estimated for each species pair serving as a proxy for trophic and prey overlap.

PERMANOVA analyses were performed using PERMANOVA+, the add-on for PRIMER-E v6 [59], and standard ellipses were estimated with the SIBER function in the SIAR v. 4.2 R package [58]. All other statistical analyses were conducted in R [60].

## 2.5. Stable isotope mixing models

Bayesian isotope mixing models were employed to identify the most probable feeding areas of blue, fin and sei whales during winter and early spring. The potential range of these whales was divided into general regions based on knowledge of their distribution in the North Atlantic and of the movements of whales migrating through the Azores (for details, see electronic supplementary material). These regions were: Northwest Atlantic (NWA), including the Gulf of St. Lawrence, Newfoundland, western and southwestern Greenland; Northeast Atlantic (NEA), extending from southeastern Greenland to the Barents Sea; Iberia (IB); Bay of Biscay (BB); western Mediterranean (MED); Azores (AZ); pelagic waters off Northwest Africa (NAF), from Cape Verde to Liberia and offshore to the Guinea Dome; and the upwelling region along the Northwest Africa coast (NAF-UPW), from Morocco to Senegal (figure 1).

We identified stable isotope ratios of potential prey taxa within these regions from the literature and from data repositories (PANGEA) (figure 1; electronic supplementary material, table S1). In the North Atlantic, the bulk of sei whale diet comprises calanoid copepods, whereas blue and fin whales feed mainly on locally abundant euphausiids [61,62]. Therefore, all prey taxa from the families Calanidae and Euphausiidae were combined into two prey groups—copepods and euphausiids—which were compared against sei, and against blue and fin whale isotopic ratios, respectively. Years of prey collection did not match all the years of collection of whale skin. Thus, prey data from as many years as possible within the time frame of whale sampling were gathered to capture possible inter-annual variability in zooplankton isotopic composition. Zooplankton $\delta^{15}N$ and $\delta^{13}C$ values can also show significant seasonal variations at regional scales [27]. We limited the regional prey isotope values to the periods of the year when whales were expected to occur in that region; i.e. for NWA and NEA we included prey isotopic data from summer, for the remaining regions we used prey isotopic data from autumn, winter and spring.

We only included prey isotopic data from studies that accounted for the effect of lipids on $^{13}C$ concentrations (by extracting lipids or mathematically correcting $\delta^{13}C$ values), and that included both $\delta^{15}N$ and $\delta^{13}C$ values, measures of isotope variability and sample sizes used. For each region, prey group and season, we estimated the mean (and standard deviation, ±s.d.) prey isotopic value by summing the means and variances of all the available data, weighted by sample size. These values were used as input parameters in mixing models.

Stable isotope mixing models were run in R using the SIMMR v. 0.3 package [63]. Parameters included in the models were the $\delta^{15}N$ and $\delta^{13}C$ values of all whales (raw data) and of prey sources (entered as mean ± s.d.), and a diet–skin discrimination factor estimated for fin whales ($2.82 \pm 0.30\%$₀ for $\delta^{15}N$ and $1.28 \pm 0.38\%$₀ for $\delta^{13}C$) [64]. We performed a separate model for each whale species, including regions that whales might have exploited during the three months prior to sampling as potential sources (electronic supplementary material, table S2). Sei whales were divided into two groups based on the sampling season (spring and summer) because of the different $\delta^{13}C$ values observed in individuals sampled during spring and summer (see results below). Fin and blue whales were each treated as a single group. Fin whale models were performed excluding whales sampled during autumn.

# 3. Results

A total of 95 skin samples were analysed from blue ($n = 17$), fin ($n = 42$) and sei ($n = 36$) whales (table 1). Over 75% of the samples were collected in spring (March–May, $n = 72$). The remaining samples were

**Table 1.** Mean (±s.d.) $\delta^{15}N$ and $\delta^{13}C$ values from the skin and faeces of blue, fin and sei whales sampled off the Azores by sampling season.

| tissue | season | blue whale | | | fin whale | | | sei whale | | |
|---|---|---|---|---|---|---|---|---|---|---|
| | | $n$ | $\delta^{15}N$ | $\delta^{13}C$ | $n$ | $\delta^{15}N$ | $\delta^{13}C$ | $n$ | $\delta^{15}N$ | $\delta^{13}C$ |
| skin | winter | 2 | 9.13 ± 0.05 | −19.22 ± 0.42 | | | | | | |
| | spring | 15 | 9.10 ± 0.77 | −18.61 ± 1.06 | 34 | 9.40 ± 0.54 | −19.40 ± 0.79 | 23 | 8.95 ± 0.73 | −17.86 ± 0.73 |
| | summer | | | | 6 | 9.63 ± 0.69 | −19.68 ± 0.55 | 11 | 9.10 ± 0.37 | −16.42 ± 0.42 |
| | autumn | | | | 2 | 11.67 ± 0.36 | −19.25 ± 0.80 | 2 | 9.27 ± 0.28 | −19.24 ± 0.03 |
| | total | 17 | 9.10 ± 0.72 | −18.68 ± 1.02 | 42 | 9.54 ± 0.73 | −19.44 ± 0.76 | 36 | 9.01 ± 0.62 | −17.50 ± 1.01 |
| faeces | spring | 1 | 5.30 | −19.78 | 5 | 6.29 ± 1.02 | −20.84 ± 0.42 | 2 | 4.78 ± 1.33 | −20.68 ± 1.27 |
| | summer | | | | 5 | 6.02 ± 0.55 | −20.38 ± 0.15 | | | |
| | total | 1 | 5.30 | −19.78 | 10 | 6.16 ± 0.78 | −20.61 ± 0.38 | 2 | 4.78 ± 1.33 | −20.68 ± 1.27 |

**Table 2.** Results of one-way PERMANOVA analyses for differences in isotopic ratios ($\delta^{15}$N and $\delta^{13}$C values) among whale species, and among sexes, seasons and years within whale species. SS: sums of squares; d.f.: degrees of freedom; pseudo-$F$: $F$-values by permutation; $P$: permutational probability values based on Monte Carlo random draws.

| analysis | sources of variation | SS | d.f. | pseudo-$F$ | $P$ |
|---|---|---|---|---|---|
| inter-species | species | 57.46 | 2 | 20.247 | <0.001 |
| | residuals | 130.54 | 92 | | |
| | total | 188.00 | 94 | | |
| blue whale | sex | 2.36 | 1 | 1.718 | 0.208 |
| | season | 1.42 | 1 | 1.030 | 0.371 |
| | year | 8.82 | 5 | 1.282 | 0.307 |
| | residuals | 12.37 | 9 | | |
| | total | 26.28 | 16 | | |
| fin whale | sex | 1.99 | 1 | 4.269 | 0.028 |
| | season | 0.34 | 1 | 0.337 | 0.464 |
| | year | 19.21 | 5 | 8.251 | <0.001 |
| | residuals | 15.36 | 33 | | |
| | total | 56.13 | 41 | | |
| sei whale | sex | 0.01 | 1 | 0.01 | 0.814 |
| | season | 8.90 | 2 | 4.449 | <0.001 |
| | year | 19.42 | 6 | 3.237 | <0.001 |
| | residuals | 12.71 | 26 | | |
| | total | 48.13 | 35 | | |

collected in summer (June–August, $n = 17$), autumn (September–November, $n = 4$) and winter (December–February, $n = 2$). No deviations from parity in sex ratios were observed in any whale species ($\chi^2$-test, $p > 0.05$). Additionally, 13 faecal samples were collected from blue ($n = 1$), fin ($n = 10$) and sei ($n = 2$) whales (table 1).

## 3.1. Intra-specific variability in isotopic ratios

The stable isotope ratios of blue whale skin ranged from –19.9‰ to –16.2‰ for $\delta^{13}$C values and from 8.0‰ to 11.0‰ for $\delta^{15}$N values. Fin whale skin ranged from –20.4‰ to –17.0‰ in $\delta^{13}$C values and from 8.6‰ to 11.9‰ in $\delta^{15}$N values, while sei whale skin ranged from –19.3‰ to –15.9‰ in $\delta^{13}$C values and from 7.3‰ to 10.3‰ in $\delta^{15}$N values. All whale species displayed slightly higher variability in $\delta^{13}$C values than in $\delta^{15}$N values (table 1). In blue whales, none of the factors analysed—sex, season and year—explained the variability in the isotope ratios (table 2) and univariate analyses were not carried out.

No significant effect of season on fin whale isotope ratios was detected (table 2). However, because the two fin whales sampled during autumn showed higher $\delta^{15}$N values (greater than 2‰) relative to fin whales sampled during the spring and summer (table 1), univariate models were conducted excluding isotope data from autumn to avoid possible confounding effects of season and year. Fin whale isotopic ratios varied significantly among years (electronic supplementary material, table S3). Fin whales sampled in 2008 showed higher $\delta^{15}$N values than whales sampled in other years (figure 2). Whales sampled in 2008 had higher $\delta^{13}$C values than those from 2013 and 2014, and samples from 2010 showed higher $\delta^{13}$C than those from 2014 (figure 2). While the PERMANOVA indicated a significant effect of sex, the univariate tests did not detect significant differences in $\delta^{15}$N and $\delta^{13}$C values between males and females (electronic supplementary material, table S3).

In contrast to fin whales, sei whale isotopic ratios varied mainly among seasons and to a lesser extent among years, with no significant effect of sex (table 2). Univariate analysis showed that seasonal differences in $\delta^{15}$N values were not statistically significant but $\delta^{15}$N values differed significantly among years (electronic supplementary material, table S3). In 2014, sei whales were more depleted in $^{15}$N than whales sampled in 2005, 2008 and 2009 (figure 2). Season explained nearly 54% of the

**Figure 2.** Annual variability in $\delta^{15}N$ (top panel) and $\delta^{13}C$ (bottom panel) values in the skin of blue, fin and sei whales. The black dots and thick black lines correspond to the mean and median, respectively; the box edges are the 25th and 75th percentiles, the whiskers are the extremes and the open circles represent outliers in the data. Sample sizes are given above the boxplots. Only years with greater than or equal to two samples used in statistical analyses are shown. Inter-annual differences in blue whale $\delta^{15}N$ and $\delta^{13}C$ values were not statistically significant. Fin whale boxplots do not include isotope values from autumn.

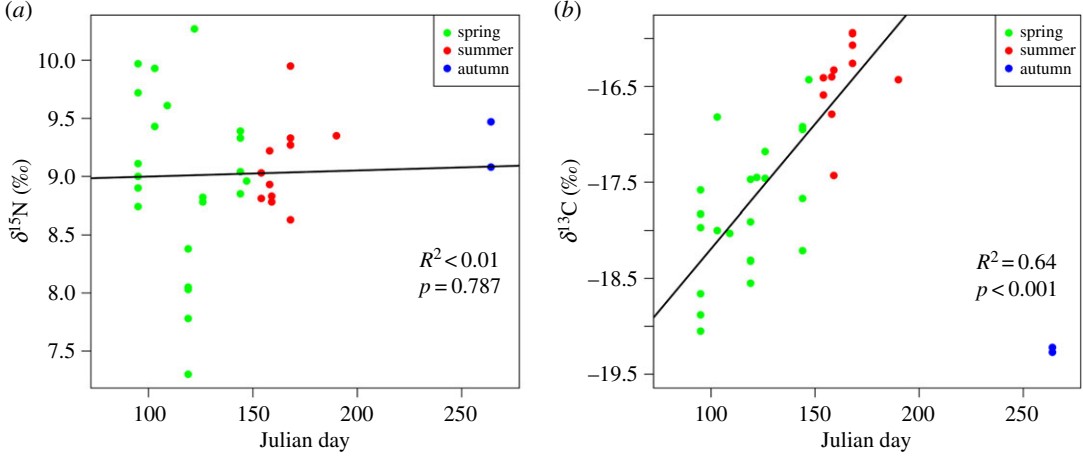

**Figure 3.** Relationship of isotopic ratios with sampling date for sei whales. The black solid line represents the fit of the linear regression model between $\delta^{15}N$ (a) or $\delta^{13}C$ (b) values in the skin of sei whales and sampling date, expressed in Julian days. Models were fitted only to isotopic data from spring (green circles) and summer (red circles) and values for autumn (blue circles) are included only for illustrative purposes. The relationship between $\delta^{15}N$ values and Julian day was not significant.

variance in $\delta^{13}C$ values in sei whale skin, while inter-annual differences accounted for 21% (electronic supplementary material, table S3). The highest $\delta^{13}C$ values were measured in sei whales sampled in summer and the lowest values in whales sampled in autumn, whereas animals from spring displayed intermediate values. Moreover, $\delta^{13}C$ values increased with sampling day for sei whales sampled during spring and summer ($R^2 = 0.64$, $p < 0.001$), such that whales sampled in mid-summer had $\delta^{13}C$ values 2.6‰ higher than those sampled in mid-spring (figure 3). The increase in $\delta^{13}C$ value was independent from the sex of the whales, as males and females showed similar slopes and $y$–intercepts of the regression line. No relationship was detected between $\delta^{15}N$ values and sampling day ($R^2 = 0.002$, $p = 0.787$) (figure 3).

## 3.2. Inter-specific differences in isotopic ratios

Blue, fin and sei whale skin showed distinct isotopic ratios (electronic supplementary material, table S3). Univariate tests followed by pairwise comparisons indicated that sei whales (–17.5 ± 1.01‰), blue whales

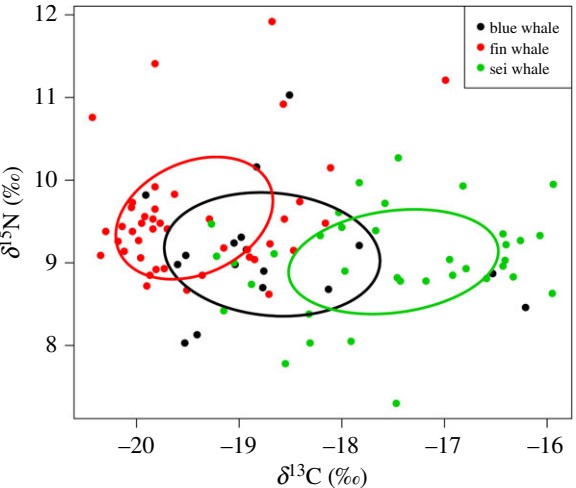

**Figure 4.** $\delta^{13}$C and $\delta^{15}$N biplot in the skin of blue ($n = 17$), fin ($n = 42$) and sei ($n = 36$) whales sampled off the Azores. Solid line represents the standard ellipse areas (SEAc) estimated for each whale species.

($-18.7 \pm 1.02‰$) and fin whales ($-19.4 \pm 0.76‰$) were all significantly different from each other in $\delta^{13}$C values. Only fin and sei whales showed significant differences in $\delta^{15}$N values, with fin whales presenting higher values ($9.5 \pm 0.73‰$) than sei whales ($9.01 \pm 0.62‰$).

Blue whales had the largest estimated $SEA_C$ and fin whales the smallest $SEA_C$ (figure 4). There was no overlap in $SEA_C$ between fin and sei whales but both species shared their niche space with blue whales (43% of fin whale $SEA_C$ and 34% of sei whale $SEA_C$ overlapped with blue whale $SEA_C$). Species differences in isotopic space were mostly in $\delta^{13}$C values although $\delta^{15}$N values also contributed to the separation of $SEA_C$ of blue and fin whales.

With respect to isotopic ratios in whale faeces, fin whale faeces had higher $\delta^{15}$N values ($6.2 \pm 0.78‰$) than those of sei whales ($4.8 \pm 1.33‰$) but overlapped those of blue whales ($5.3‰$). In contrast, faecal $\delta^{13}$C values were similar in all three species (table 1). Results from paired faecal and skin samples of two fin whales showed lower isotopic ratios in faeces than in skin: by 3.65‰ and 3.81‰ for $\delta^{15}$N values and by 0.46‰ and 0.81‰ for $\delta^{13}$C values.

## 3.3. Prey isotopic composition by region and season

Copepods and euphausiids sampled during summer in the NWA and NEA were highly $^{15}$N-enriched and $^{13}$C-depleted relative to prey from other regions and formed a distinct cluster (electronic supplementary material, figure S1). Isotopic ratios of prey from the remaining regions also formed relatively distinct clusters, but there was considerable within-region variability in the $\delta^{15}$N values of prey sampled during the spring and autumn (electronic supplementary material, figure S1). Overall, the highest $\delta^{13}$C and the lowest $\delta^{15}$N values of copepods and euphausiids were observed in the NAF-UPW and NAF regions, although there was some overlap with other regions. Isotopic ratios of prey from the AZ, IB, BB and MED were intermediate between those of NWA/NEA and NAF-UPW/NAF. In the NAF-UPW and NAF, spring copepods and euphausiids had significantly higher $\delta^{15}$N values than those from autumn, while the opposite pattern was detected in euphausiids from IB.

## 3.4. Stable isotope mixing models

Skin $\delta^{15}$N and $\delta^{13}$C values of blue, fin and sei whales were outside the range of isotopic values of prey (adjusted to account for trophic discrimination) from NWA and NEA (figure 5). In contrast, whale isotopic ratios were generally bounded by prey from temperate to tropical regions, indicating these regions as plausible feeding areas (figure 5).

Mixing model results indicate that blue whales mainly used food resources available in NAF and NAF-UPW during autumn, and in the AZ during spring (figure 6a; electronic supplementary material, table S4). However, the model was poorly resolved as the 95% credibility intervals overlapped substantially across nearly all sources. Fin whale skin primarily reflected euphausiids from the IB during spring (64%, 95% CI:

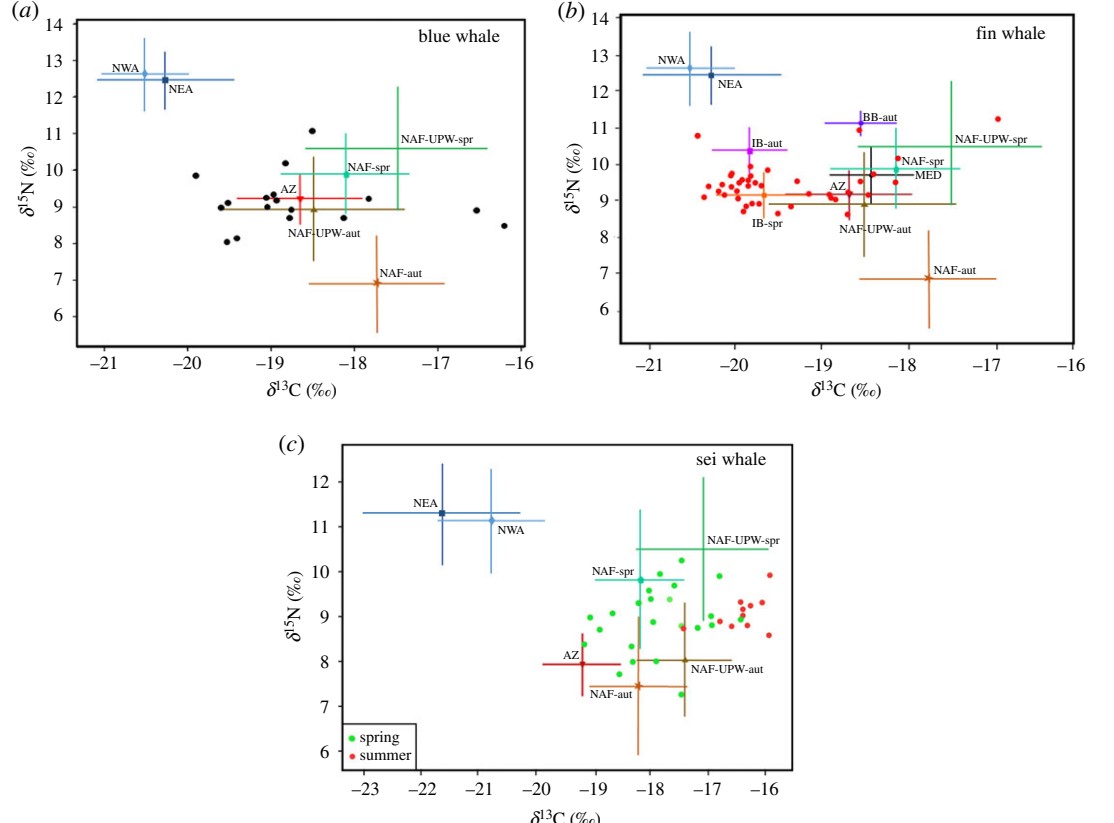

**Figure 5.** Isotopic biplots of blue (*a*), fin (*b*) and sei (*c*) whales in relation to potential prey sources from different regions within the North Atlantic. $\delta^{15}N$ and $\delta^{13}C$ values of prey are shown as means ± s.d. Prey isotopic ratios were adjusted for trophic discrimination values [64]. Regions are: AZ, Azores; IB, Iberia; BB, Bay of Biscay; NAF, North Africa; NAF-UPW, North Africa upwelling; NEA, Northeast Atlantic; NWA, Northwest Atlantic; MED, Mediterranean. NAF and NAF-UPW include prey sampled during spring (spr) and autumn (aut).

37–81%), with the remaining sources contributing negligible amounts (3–5%) to fin whale diet (figure 6*b*; electronic supplementary material, table S4). Sei whale mainly used copepods from NAF-UPW, with a lower contribution from NAF and AZ (figure 6*c,d*; electronic supplementary material, table S4). The relative contribution of these regions varied slightly between sei whales sampled in spring and summer. Sei whales sampled in spring exploited proportionally more the NAF-UPW region during autumn, while those sampled in summer showed increased use of NAF-UPW during spring, as well as of NAF and AZ.

# 4. Discussion

This study documents $\delta^{13}C$ and $\delta^{15}N$ values in the skin and faeces of blue, fin and sei whales encountered in temperate waters around the Azores. Mixing model results revealed little dietary contribution from high-latitude summer feeding grounds to the diet of the three whale species but were consistent with the notion of feeding in subtropical and tropical regions during the winter and spring. Moreover, blue, fin and sei whales occupied distinct isotopic spaces and the isotopic distance among species was mainly driven by differences in $\delta^{13}C$ values, suggesting that the species use different winter and spring habitats. The latter observation was supported by the isotopic models that indicate distinct probable feeding regions for each whale species.

## 4.1. Intra-specific variability in isotopic ratios

There was no evidence of annual or seasonal differences in isotope ratio for blue whales sampled in the Azores, probably because of the small number of samples analysed. However, we observed large differences in skin $\delta^{13}C$ values (1.6–3.4‰) among blue whales sampled during the same year and month that were not matched by differences in $\delta^{15}N$ values. Likewise, the temporal variation in skin

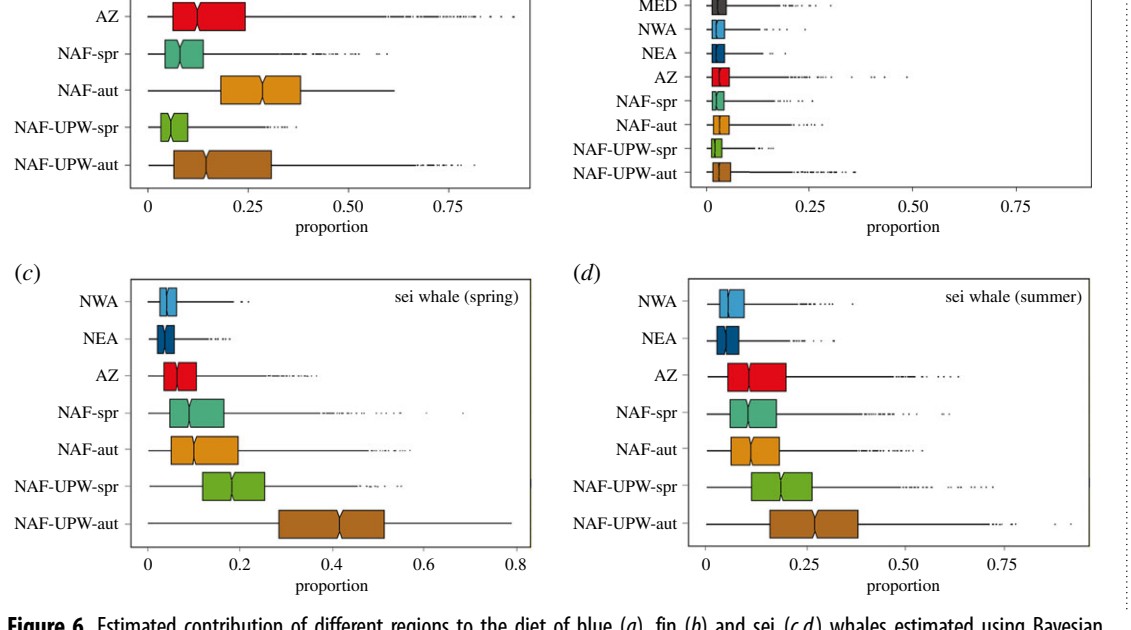

**Figure 6.** Estimated contribution of different regions to the diet of blue (*a*), fin (*b*) and sei (*c,d*) whales estimated using Bayesian isotopic mixing models. Results show posterior model estimates (median, interquartile range and max/min values) of source contribution to whale skin tissue. Regions are: AZ, Azores; IB, Iberia; BB, Bay of Biscay; NAF, North Africa; NAF-UPW, North Africa upwelling; NEA, Northeast Atlantic; NWA, Northwest Atlantic; MED, Mediterranean. NAF and NAF-UPW include prey sampled during spring (spr) and autumn (aut).

isotopic ratios observed in fin and sei whales was higher for $\delta^{13}C$ than for $\delta^{15}N$ values, and changes in $\delta^{13}C$ values were decoupled from variations in $\delta^{15}N$ values. These findings indicate that differences in diet may not be the main explanation for the isotopic variation in whale species, and instead suggest that whales foraged in spatially and/or temporally distinct food webs.

The high variability observed in $\delta^{13}C$ values among blue whales sampled just a few days apart suggests that individuals within the population foraged across a strong gradient in baseline $\delta^{13}C$ values. Such gradients are often associated with sharp transitions in biogeochemistry, oceanography and productivity across major biogeographic domains [27,65,66].

Although statistically significant, inter-annual differences in fin whale isotope ratios should be treated with caution due to small sample sizes in most years. Additionally, owing to the lack of prey samples from all fin whale sampling years, annual variations in skin $\delta^{13}C$ and $\delta^{15}N$ values could not be related to those of potential prey. Nevertheless, several different hypotheses could explain the decrease in fin whale $\delta^{13}C$ values over the years. Long-term changes in sea-surface temperature, upwelling intensity and zooplankton composition documented off the Iberian coast [67,68] (the most probable feeding region of fin whales identified in the mixing models) could affect baseline $\delta^{13}C$ values and ultimately those of fin whales. However, these oceanographic and biological changes were evident at decadal scales. Fluctuations between successive years were small and unlikely to cause significant trends in baseline $\delta^{13}C$ values. Similarly, the estimated rate of change in $\delta^{13}C$ of dissolved inorganic carbon (DIC) in the North Atlantic attributable to the Suess effect (the decrease in the $\delta^{13}C$ values of atmospheric $CO_2$ from the combustion of fossil fuels) ranges between $-0.007‰$ per year in polar regions and $-0.025‰$ per year in subtropical regions [69–71]. This value is too low to account for the annual differences in fin whale $\delta^{13}C$ values. The lack of seasonal differences in fin whale isotopic ratios also indicates that time of sampling was unlikely to be the cause of the observed inter-annual variation. Hence, a change in foraging location, towards more [13]C-depleted regions, appears to be the most parsimonious explanation for the observed decline in $\delta^{13}C$ values in fin whales.

Sei whale isotopic ratios varied little between years and mostly with season. The most evident pattern was a progressive increase in $\delta^{13}C$ values with sampling date for whales sampled during the spring and summer, while the two individuals sampled in autumn had the lowest $\delta^{13}C$ values. Differences in $\delta^{13}C$ values between whales sampled in spring and summer may partly reflect the seasonality in baseline values in the NAF-UPW

and NAF regions, identified as the most likely foraging areas for these sei whales. In these regions, $\delta^{13}$C and $\delta^{15}$N values are expected to increase during the period from winter to late spring, following the increase in phytoplankton production and the concomitant decrease in nitrate ($^{14}$N) concentration (due to the preferential uptake of the lighter isotope by phytoplankton) during the development of upwelling-induced and spring blooms [35,72,73]. As whale skin presumably reflects the isotopic composition of prey consumed during the three to four months preceding sampling, sei whales sampled during spring were expected to have lower $\delta^{13}$C and $\delta^{15}$N values than individuals sampled during summer. There may be several explanations for the lack of differences in $\delta^{15}$N values: foraging occurred before the exhaustion of the $^{14}$N pool and the subsequent increase in $\delta^{15}$N values in phytoplankton; changes in $\delta^{15}$N values at the food-web base were not immediately reflected in copepods owing to their longer turnover times [74]; and longer incorporation time for $\delta^{15}$N than $\delta^{13}$C values in whale skin tissue [48], because of the distinct metabolic pathways of these elements [47]. Seasonal patterns in sei whale isotopic values may also reflect foraging across regions with distinct isotopic baselines and different temporal isotopic dynamics. Future studies using other chemical tracers or individual amino acids could help to elucidate the underlying causes for the observed seasonal variation.

The two sei whales sampled in late September had the lowest $\delta^{13}$C values recorded for the species. At the time of sampling, these two whales seemed to be migrating to the wintering habitats, as indicated by the satellite tracking data from one of these whales that was heading towards the Canary islands when the tag stopped transmitting [22]. Hence, we assume the isotopic ratios in the skin of sei whales sampled in autumn reflect feeding at higher latitudes.

We did not find differences in stable isotope values between male and female blue, fin and sei whales. Differences in trophic niche between male and female baleen whales observed outside the breeding season have been attributed to spatial segregation and dietary differences between the sexes, possibly reflecting intra-specific competition and individual or social preferences [42]. Male and female baleen whales are expected to co-occur in the same ranges during the winter breeding/mating period, and possible differences in habitat selection at fine scales may not be reflected in the isotopic composition of their skin. Sex differences in diet could also result from the different nutritional requirements of males and females; yet, the reproductive status of individuals appears to have a stronger influence in energy demands than sex [75].

## 4.2. Inter-specific differences in isotopic ratios

Despite the intra-specific variability in isotopic ratios observed in blue, fin and sei whales, the three species occupied distinct isotopic spaces and showed different mean isotopic ratios. Fin whale skin had higher $\delta^{15}$N values than that of sei (0.53‰) and blue whales (0.44‰), although the differences from blue whales were not statistically significant. $\delta^{15}$N values mainly reflect the trophic position of consumers [32]. Hence, these results suggested a greater contribution of higher-trophic level organisms to the diet of fin whales, consistent with knowledge on feeding habits of these whale species. The diet of North Atlantic fin whales is dominated by euphausiids (especially *Meganyctiphanes norvegica* and *Thysanoessa raschii*) but they also consume a variety of small epipelagic and mesopelagic schooling fish [61,62]. Blue whales are considered dietary specialists, feeding almost exclusively on krill (*M. norvegica*, *T. raschii* and *Thysanoessa inermis*), whereas sei whales feed primarily on calanoid copepods and to a lesser extent on euphausiids [61,62]. A comparison of isotopic values of calanoid copepods and euphausiids compiled from the literature shows that, with a single exception (NAF in autumn), mean $\delta^{15}$N values of copepods were consistently lower (by 0.2–1.42‰) than those of euphausiids sampled in the same region and season. Thus, the higher reliance of sei whales on copepods would explain their lower $\delta^{15}$N values relative to fin and blue whales.

Despite these differences, there was a remarkable overlap in nitrogen isotope ratios in fin, blue and sei whales, where most individuals clustered within a narrow range (approx. 1.5‰) of $\delta^{15}$N values. This was in clear contrast with studies from feeding grounds in the Gulf of St. Lawrence where $\delta^{15}$N values in fin whales were found to be 2‰ higher than those in blue whales [42]. The small differences in $\delta^{15}$N values observed among whale species in our study may partly be explained by exploitation of distinct isotope baselines, diluting variability in $\delta^{15}$N values from dietary differences.

Fin whale skin had the lowest $\delta^{13}$C values, on average approximately 2‰ and approximately 1‰ lower than values detected in sei and blue whales, respectively. The low $\delta^{13}$C values in fin whales indicate a majority of carbon derived from pelagic phytoplankton-based food webs or feeding at high-intermediate latitudes [27,36,72]. Conversely, the higher $\delta^{13}$C values seen in sei whales suggest increased foraging in systems characterized by relatively high productivity. Blue whales showed the largest range in $\delta^{13}$C values and a mean $\delta^{13}$C that was intermediate between that of fin and sei

whales. Taken together these results indicate that blue whales use a wider range of isotopically distinct habitats than the other two species.

## 4.3. Evidence of feeding during winter migration

The regional $\delta^{15}N$ and $\delta^{13}C$ values of copepods and euphausiids compiled from the literature agreed well with broad geographical patterns of both isotopes in the North Atlantic Ocean, and were consistent with latitudinal and longitudinal trends in oceanic productivity and nutrient sources [27,72].

The skin isotopic composition in blue, fin and sei whales sampled in the Azores did not reflect that of prey consumed during summer in the NWA and NEA foraging grounds. This conclusion is supported by the results of the mixing models, which detected a minimal contribution of prey from these areas to the isotopic ratios of whales. Our results indicate that sei whales recently foraged in food webs that are considerably depleted in $^{15}N$ and enriched in $^{13}C$ compared with copepods from the NWA and NEA. Blue and fin whales foraged on euphausiids with lower $\delta^{15}N$ and slightly higher $\delta^{13}C$ values than those found during the summer in NWA and NEA. Such isotopic ratios are characteristic of low-latitude food webs. The outcome of the mixing model assessments suggests that whale isotope ratios primarily reflected prey from subtropical and tropical regions, and clearly demonstrates that blue, fin and sei whales fed during the winter migration.

We could not find information on isotope ratios of North Atlantic sei whales to compare with our results but the isotopic composition of blue and fin whales sampled at higher latitude habitats supports these findings. The mean $\delta^{15}N$ values in fin whales sampled off the Azores were lower than values reported in fin whales sampled in the Gulf of St. Lawrence (–2.86‰) [42] and Celtic Sea (–2.36 ‰) [54]. The same was the case for blue whales sampled off the Azores when compared with values from the Gulf of St. Lawrence [42], although the difference was less pronounced than for fin whales. Fin whales from the Azores were also slightly depleted in $^{13}C$ relative to individuals from higher latitudes (St. Lawrence: –0.84‰, Celtic Sea: –1.24‰), whereas blue whales had values similar to those estimated in St. Lawrence blue whales. While this pattern contradicts the negative trend in baseline and prey $\delta^{13}C$ values with latitude [76] it is consistent with the hypothesis that St. Lawrence and Celtic whales may feed in high-nutrient $^{13}C$-rich inshore systems [42,54].

## 4.4. Winter and spring habitat use

The use of Bayesian mixing models enabled an assessment of the relative contribution of different regional sources to the diet of baleen whales. Yet, the results and discriminatory power of mixing models strongly depend on the number of prey sources and the isotopic separation among them, and can be biased by missing prey sources [77]. In blue and fin whales, skin samples with the lowest and highest $\delta^{13}C$ values were not well bounded by regional isotopic values. In the case of sei whales, this affected only the summer samples (although all values were within the standard deviation of prey $\delta^{13}C$ values). This implies that either prey from important foraging regions was missing or prey from the regions included were not adequately represented. Although we cannot dismiss the first hypothesis, prey isotopic data from tropical and subtropical regions were limited to a few taxa and years, and are unlikely to represent the range of prey items consumed by whales and their annual isotopic variability. More importantly, we did not have access to prey isotopic data from the winter, which presumably is reflected in the isotopic composition of whales sampled during spring. Additionally, by combining prey taxa into groups and broad geographical regions, we increased the amount of isotopic variation of regional prey sources and the degree of overlap between them. In conclusion, the results obtained from the mixing model assessment might change with additional samples from these and possibly other regions. Nevertheless, these results help shed light on the large-scale winter habitat use of each whale species.

The model results indicate that blue and sei whales foraged primarily in food webs supported by the NAF-UPW system, as well as in less productive pelagic food webs within tropical and subtropical regions. The majority of blue whales showed a $\delta^{13}C$ value consistent with foraging on open-ocean phytoplankton sources. The relatively high dietary contribution of the NAF-UPW to blue whales was explained by the two whales with a highly enriched $^{13}C$. Although these individuals were clear outliers in our sample, whaling records and historic and contemporary sightings support the occurrence of blue whales in the upwelling region off Mauritania in winter [78–80]. Prey from NAF-UPW dominated the diet of sei whales sampled during the spring. In summer, the dietary contribution of NAF-UPW decreased and that of lower productivity regions (NAF and AZ)

increased. This is in clear contradiction to the increasing trend in $\delta^{13}C$ observed from spring to summer samples and is probably an artefact from the lack of prey sources with $\delta^{13}C$ values as high as those of sei whales. Use of the Northwest African upwelling (from Morocco down to Senegal) by sei whales is well documented in the literature [78–80] and a recent winter sighting survey conducted southwest of Cap Blanc, Mauritania, reported large aggregations of sei whales feeding over the continental shelf and shelf-break [67]. We suggest that these whales exploited the nutrient-rich upwelling off Northwest Africa to take advantage of the high abundance and biomass of zooplankton [81].

The results of the fin whale model indicate an overwhelming predominance of the Iberia region to fin whale diet, probably because of the influence of the dense cluster of low $\delta^{13}C$ values, composed mainly of samples from 2014. As fin whale $\delta^{13}C$ values varied significantly among years, this implies that model results only represent regional contributions to fin whale diet in 2014. Nevertheless, these findings suggest a previously undocumented migratory link between the Northeast Atlantic and the Central Atlantic. It remains unclear if the connectivity between the two regions detected only in 2014 represents a recent shift in fin whale movement patterns. Finally, model results corroborate our previous interpretation of isotopic ratios and clearly show that fin whales did not feed in the NAF-UPW, in agreement with the scarcity of sightings in the area [67].

## 4.5. Effects of nutritional restriction on skin isotope ratios

Physiological changes associated with periods of fasting or nutritional restriction can affect stable isotope dynamics, and hence the isotopic ratios in different animal tissues, complicating interpretation of isotopic analyses. The most commonly reported effect is an increase in $\delta^{15}N$ (usually of less than 1‰) in plasma and newly formed tissues, as a consequence of protein synthesis using $^{15}N$-enriched amino acids derived from catabolism of endogenous protein [82]. Although $\delta^{13}C$ appears less sensitive to nutritional stress [82], some studies documented a decrease in plasma $\delta^{13}C$ values in hibernating bears, as a result of the mobilization of lipid stores during fasting [83].

Contrary to expectations, the $\delta^{15}N$ values in the baleen plates of fin whales and both $\delta^{15}N$ and $\delta^{13}C$ values in baleen of bowhead whales (*Balaena mysticetus*) increased during the summer, a period of intensive feeding, and declined during the winter, when whales were assumed to feed less [44,84,85]. The lack of $\delta^{15}N$ increase was attributed to the whale's ability to maintain a positive nitrogen balance through limited winter feeding, reduced excretion rates and reliance on lipid catabolism to fulfil energetic and physiological needs, thereby avoiding protein mobilization [84]. Under these circumstances, variations in baleen $\delta^{15}N$ values would reflect variation in baselines between summer and winter grounds [84].

Lipids and carbohydrates contain negligible amounts of nitrogen [86], and skin maintenance and regeneration require a protein source. The close agreement between whale skin and lower latitude prey $\delta^{15}N$ values and the high discrepancy to $\delta^{15}N$ values of prey and conspecifics from higher latitudes suggest that nitrogen in whale skin was derived mostly from dietary rather than endogenous sources, even during periods of food restriction. On the other hand, non-essential amino acids that are major components of skin tissue can be synthesized from lipid-derived carbons [87] that could result from the mobilization of blubber stores. Considering that lipid reserves of whales are accumulated during the intensive summer feeding period [7] and that lipids are $^{13}C$-depleted relative to proteins [29], incorporation of lipid-derived carbon into skin tissue should result in $\delta^{13}C$ values lower than those of prey from northern foraging grounds. Again, our results contradict these predictions and indicate that skin $\delta^{13}C$ values reflected recently consumed prey. Overall, these findings show that $\delta^{15}N$ and $\delta^{13}C$ values in blue, fin and sei whale skin were not driven by physiological changes associated with periods of food restriction.

## 4.6. Isotopic ratios in faeces

Analyses of faeces are indicative of recent diet [40] and we assume they represent prey consumed in the pelagic waters in or around the Azores. As expected, $\delta^{15}N$ values in whale faeces were slightly lower (0.1–1.1‰) than those in euphausiids and copepods collected at the Azores, probably because of selective removal of lighter nitrogen during digestion and assimilation [88]. Similarly to what was observed in skin tissue, fin whale faeces had the highest $\delta^{15}N$ value and those of sei whales the lowest, indicating that sei whales feed at a lower trophic level than fin and blue whales. Unsurprisingly, $\delta^{13}C$ values were similar among all species and agreed well with prey values from the regional pelagic food webs.

# 5. Conclusion

Our work provides strong evidence that North Atlantic blue, fin and sei whales migrating through central Atlantic waters feed during the winter or early spring in tropical and subtropical waters, and that this strategy is more prevalent among migratory whales than is currently accepted. Although the significance of winter feeding to the overall energetic budget of the animals remains unclear, these results contribute to a growing body of evidence suggesting that baleen whale migration is much more complex than previously assumed and cannot be reduced to a simple 'feast and famine' paradigm.

Our findings support a distributional range for sei whales spanning across the North Atlantic basin. All sei whales (n = 7) instrumented with satellite tags in the Azores during spring and early summer (April–June) migrated to the Labrador Sea [22,89]. Isotopic measurements were available for five of these whales which showed a $\delta^{13}$C increased value, similar to the majority of sei whales analysed. Together these results showed that part of the sei whale population feeding in the Labrador Sea in the summer most likely originate from the wintering grounds off the Northwest African coast, as proposed earlier [22].

Finally, given projected changes in the upwelling system off Northwest Africa under climate change [90], the possible reliance of sei whales on food webs from this region during winter is of concern and deserves to be further investigated.

SIA is increasingly used to examine trophic ecology and movements of baleen whales. These studies are generally based on isotopic measurements along the length of baleen plates, a tissue that is metabolically inert after synthesis and therefore can retain long-term records of feeding patterns. Collection of baleen plates depends on the availability of stranded specimens, making it difficult to obtain sufficient numbers of samples to document present–day movement patterns at the population level. We believe that the analysis of skin from biopsy samples collected from multiple times of the year and locations across the populations' range offers an alternative to analysis of baleen plates. Still, employing stable isotopes to track migration and diet of baleen whales will require a better understanding of the factors affecting isotopic incorporation and trophic discrimination between whale tissues and their diet, as well as more detailed marine isoscapes and prey isotope ratios.

Ethics. Fieldwork and sample collection were approved and conducted under scientific permits from the Regional Directorate for the Environment, Regional Government of the Azores (7/ 2005, 4/2006, 76/2007, 20/2009, 16/2010, 51/2011, 31/2012, 20/2013, 34/2014). All procedures followed the guidelines of the American Society of Mammalogists [91].
Data accessibility. The dataset supporting this article has been uploaded as part of the electronic supplementary material.
Authors' contributions. M.A.S. conceived, designed and coordinated the study, secured funding, collected the samples, processed the data, conducted the statistical analyses and wrote the manuscript; A.B. conducted the stable isotope analyses and Bayesian mixing models; R.P. collected the samples; P.G. and A.C. conducted the stable isotope analyses; M.B. and P.J.P. conducted the genetic analyses. All authors helped to draft the manuscript and gave final approval for publication.
Competing interests. We declare we have no competing interests.
Funding. This work was supported by Fundação para a Ciência e Tecnologia (FCT), Azores 2020 Operational Programme and Fundo Regional da Ciência e Tecnologia (FRCT) through research projects FCT-Exploratory project (IF/00943/2013/CP1199/CT0001) and WATCH IT (Acores-01-0145-FEDER-000057) co-funded by FEDER, COMPETE, QREN, POPH, FSE, and the Portuguese Ministry for Science and Education. We also acknowledge funds provided by FCT to MARE, through the strategic project UID/MAR/04292/2019. M.A.S. (IF/00943/2013) and A.C. (IF/00029/2014/CP1230/CT0002) are supported by FCT-Investigator contracts and R.P. by an FCT postdoctoral grant no. (SFRH/BPD/108007/2015).
Acknowledgements. We thank Cláudia Oliveira, Irma Cascão, Marta Tobeña, Rebecca Boys, João Medeiros, Yves Cuenot, Pablo Chevallard Navarro, and numerous volunteers who over the years helped with data collection and organization of the photo-identification catalogue. We are also grateful to our skippers, Renato Bettencourt, Paulo Martins and Vitor Rosa. We thank three anonymous reviewers for substantially improving an earlier version of this paper.

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
