## [Reviewer comments · Royal Society Open Science]

Review History

RSOS-181800.R0 (Original submission)

Review form: Reviewer 1

Is the manuscript scientifically sound in its present form?

Yes

Are the interpretations and conclusions justified by the results?

Yes

Is the language acceptable?

Yes

Is it clear how to access all supporting data?

Yes

Do you have any ethical concerns with this paper?

No

Have you any concerns about statistical analyses in this paper?

No

Recommendation?

Accept as is

Comments to the Author(s)

This is a fascinating manuscript, that compares the $\delta^{15}\text{N}$ and $\delta^{13}\text{C}$ values of skin and fecal samples from three species of Balaenoptera migrating past the Azores islands over several years. The sample size is not large, particularly for blue whales, but large enough to support the inferences that the authors make from their data and analyses. The important findings include further support for the growing understanding that baleen whales feed far more on migration than the "classical" model of baleen whale migration indicates; and – a particularly interesting finding - that sei whales feed off Northwestern Africa in the winter.

I'm not an expert on isotope analyses, and so can't speak to the details of those analyses, although from my reading of that literature, this analysis seems appropriate. The discussion of whale migration and movement ecology is well written, accurate and appropriate. This strikes me as an important contribution to our understanding of baleen whales' movement ecology, and shows how stable isotopes can be used to improve understanding of the relationships between baleen whales' foraging ecology and movements.

My one very minor quibble is with the first sentence of the Conclusions. The authors state, "Our work provides strong evidence that North Atlantic blue, fin and sei whales migrating through central Atlantic waters feed during winter or early spring in tropical and subtropical waters, and that this is a widespread strategy among migratory whales." I'd suggest rewording to "Our work provides strong evidence that North Atlantic blue, fin and sei whales migrating through central Atlantic waters feed during winter or early spring in tropical and subtropical waters, and that this strategy is more prevalent among migratory whales than is currently accepted". Or something along those lines. My point being that we don't yet know that winter feeding is widespread amongst migratory whales, so that sentence isn't quite correct right now. I suspect that the authors are right, and the strategy is widespread, and I look forward to seeing work emulating their study, on whales in other ocean basins.

I have no real comments to make further, other than to congratulate the authors on a fine piece of work.

Review form: Reviewer 2**Is the manuscript scientifically sound in its present form?**

No

Are the interpretations and conclusions justified by the results?

No

Is the language acceptable?

Yes

Is it clear how to access all supporting data?

Yes

Do you have any ethical concerns with this paper?

No

Have you any concerns about statistical analyses in this paper?

No

Recommendation?

Major revision is needed (please make suggestions in comments)

Comments to the Author(s)

REVIEW OF

Stable isotopes reveal supplemental feeding and differences in winter habitats in blue, fin and sei whales migrating through the Azores,

This study investigate variability in isotope ratios from bulk skin tissue of blue, fin and sei whales collected near the Azores from 2002 to 2014. Authors also analysed faeces samples that were opportunistically collected during the same period of time. Prey data was obtained from published work, but data regarding the date of collection is not included anywhere in the manuscript or supplementary material. It is unknown if these prey data was collected within the same season (s) and years that whales skin samples were collected or from a different period of time. Authors used stable isotope analysis of C and N to understand some aspects of the feeding ecology of these whale species, but introduction about the application of this technique in cetaceans, and the meaning of d13C and d15N is poorly described in the introduction. There are various studies that had used stable isotope analysis in skin tissues from whales to understand variability in diet and habitat use by sex, feeding areas, maturity, etc. and identify migratory species from resident animals. There are also studies that included analysis of faeces and prey items. Introduce this information to the reader.

Author included only one paragraph describing isotope ratios, in particular, the spatial variability in primary producer values, but authors cited only one publication which refers to understand top predators movement (work of Graham et al. 2010). It seems that the authors are not reading important literature in regards to the factors driving variability in isotope ratios in plankton. At least, please cite the citations that Graham et al. 2010 included in their introduction to explain spatial variation in baseline values.

It is not clear what are the main goals of this work, e.g. if authors are trying to understand the contribution of prey items in the whales diet? Or if they are trying to understand the importance of the Azores as a feeding area for these species? Or if they are trying to identify which are the main feeding areas of migratory blue, sei and fin whales that are observed in the Azores? Or if the goal is to understand migration patterns? And whether these migratory patterns may contradict the classic migration routes for these species? Authors need to define clearly the objectives of their work and state their hypotheses.

In the methods, analysis, I also recommend to include Bayesian mixing models that are commonly used with stable isotope data. This analysis can help to ID the approximate feeding areas of these cetaceans (and discriminate others), rather than just saying that a mean value of X animal overlap with this other X mean value

Also, there are carbon and nitrogen isoscapes derived from zooplankton samples from the Atlantic, the work of McMahon et al. 2013, cited in this paper. Although these isoscapes also represent a snapshot within a specific period of time, authors can use this isoscape (in addition to the other data they included in the table S2) as a baseline to ID the main feeding areas of

cetaceans and discriminate whether the waters near the Azores represent an important feeding area for some species or not. However, data from this work are not included as reference of potential prey of the current manuscript. I do not understand why would the authors exclude this work that could be very helpful to understand habitat use of whales? if there is a good reason, explain it briefly, it could be important.

Authors need to include collection date in Table 1, and annual sample size in the legend of figure 3.

Table S1: not integrated well in the analyses nor paper discussion, it seems to be just an isolated table no connected to the manuscript.

Table S2: include collection date or at least collection year and season

Table S3, why blue whales were not included in the analyses?

Time series- Another important issue is the poorly discussed temporal variability in skin samples, which were collected in different seasons and years. Author's results, and figure 3 (that I think they are very interesting) shows that there is a high to moderate temporal variation (and significant differences between years) for both C or N in fin or sei whales, but these pattern was not well discussed. Samples from blue whales are not shown, and I do recommend to include these data in the graphs (despite that they were not significantly different and sample size was small). I also recommend to explain this temporal variation at the very beginning of the discussion (rather than discuss these results at the end), please explain what does this isotopic variation through time mean for each species? Are whale switching feeding areas through time? And/OR Are whales preying on different items? AND/Or is there any changes in the baseline values? Why blue whales is the exception? Please discuss these results and provide insights before you show and discuss results from SEAc.

For SEAc analyses, you can include results showed in figure 2 and include it in figure 5 (instead of just providing mean and SD values of those whales). This would be a stronger way to present your data (and you can eliminate figure 2). More importantly, after showing and discussing results of temporal variability in whale isotope ratios, authors could discuss that despite the observed temporal variation in isotope ratios for Sei and fin whales, SEAc results suggest that whales have different isotope niches during those 12 years. Then, you can discuss if these results are associated to differences in habitat or prey, or both.

Figure 5, because there is no date of collection of prey items, it is really hard to understand and evaluate whether whales are feeding in prey from a given site and no from another site, because the zooplankton isotope ratios can vary largely through time. So, you need to show the variability in zooplankton isotope ratios collected in different years and season (if they were collected in different year and season) and could be included in the supplementary material, and then make a case that despite this temporal variation, the spatial variability in isotope ratios is still strong between feeding areas (as presumably shown in figure 5). Then, authors could use Bayesian mixing models to ID the main areas where whales were feeding.

Results, please include subheadings, and follow the same order to discuss results. Discussion section should not be an extension of results, please concise discussion. As it is, the discussion sections are very confusing, and hard to follow.

The very first sentence of the discussion, authors claimed to be the first one measuring isotope ratios in some specific areas, why this claim is the highlight of your discussion? This is not really interesting, it is more interesting to discuss better the main results of their study, and in relation to their hypotheses that were largely missing in this manuscript.

Page 6, L10, authors need to provide the expected baseline isotope values for upwelling areas vs nearshore areas, and oligotrophic areas in the Atlantic, and include labels in the figure 5 to ID

which prey values represent oligotrophic, nearshore and upwelling areas. Otherwise, your discussion is very speculative.

General Evaluation: Because authors did not specify the goals of their work, it was really hard to follow their manuscript, and understand why they are using some analyses instead of other ones, and how one analyses connect with one specific goal. The reader has to guess most of the time. Because there are many missing pieces of information in each section of the manuscript, the study reads weak and discussion read very speculative.

Decision letter (RSOS-181800.R0)

17-Dec-2018

Dear Dr Silva,

The editors assigned to your paper ("Stable isotopes reveal supplemental feeding and differences in winter habitats in blue, fin and sei whales migrating through the Azores") have now received comments from reviewers. We would like you to revise your paper in accordance with the referee and Associate Editor suggestions which can be found below (not including confidential reports to the Editor). Please note this decision does not guarantee eventual acceptance.

Please submit a copy of your revised paper before 09-Jan-2019. Please note that the revision deadline will expire at 00.00am on this date. If we do not hear from you within this time then it will be assumed that the paper has been withdrawn. In exceptional circumstances, extensions may be possible if agreed with the Editorial Office in advance. We do not allow multiple rounds of revision so we urge you to make every effort to fully address all of the comments at this stage. If deemed necessary by the Editors, your manuscript will be sent back to one or more of the original reviewers for assessment. If the original reviewers are not available, we may invite new reviewers.

- Data accessibility

If you wish to submit your supporting data or code to Dryad (<http://datadryad.org/>), or modify your current submission to dryad, please use the following link:
<http://datadryad.org/submit?journalID=RSOS&manu=RSOS-181800>

- Competing interests

- Authors' contributions

- Acknowledgements

- Funding statement

Please note that Royal Society Open Science charge article processing charges for all new submissions that are accepted for publication. Charges will also apply to papers transferred to Royal Society Open Science from other Royal Society Publishing journals, as well as papers submitted as part of our collaboration with the Royal Society of Chemistry (<http://rsos.royalsocietypublishing.org/chemistry>). If your manuscript is newly submitted and subsequently accepted for publication, you will be asked to pay the article processing charge, unless you request a waiver and this is approved by Royal Society Publishing. You can find out more about the charges at <http://rsos.royalsocietypublishing.org/page/charges>. Should you have any queries, please contact openscience@royalsociety.org.

on behalf of Dr Asha de Vos (Associate Editor) and Kevin Padian (Subject Editor)
openscience@royalsociety.org

Associate Editor's comments (Dr Asha de Vos):

Associate Editor: 1

Comments to the Author:

Thank you for an interesting piece of work. However, reviewer 2 has made an insightful and thoughtful review that will help this manuscript in many ways. It requires new analysis and graphs etc and therefore will require extensive work which is why we are requesting a major revision. Thank you

Associate Editor: 2

Comments to the Author:

Pg 2, line 13: Arabian Humpback whales should be changed to Arabian Sea Humpback whales. You can address this once the paper comes back from peer-review.

Comments to Author:

Reviewers' Comments to Author:

Reviewer: 1

Comments to the Author(s)

This is a fascinating manuscript, that compares the $\delta^{15}\text{N}$ and $\delta^{13}\text{C}$ values of skin and fecal samples from three species of Balaenoptera migrating past the Azores islands over several years. The sample size is not large, particularly for blue whales, but large enough to support the inferences that the authors make from their data and analyses. The important findings include further support for the growing understanding that baleen whales feed far more on migration than the "classical" model of baleen whale migration indicates; and - a particularly interesting finding - that sei whales feed off Northwestern Africa in the winter.

I'm not an expert on isotope analyses, and so can't speak to the details of those analyses, although from my reading of that literature, this analysis seems appropriate. The discussion of whale migration and movement ecology is well written, accurate and appropriate. This strikes me as an important contribution to our understanding of baleen whales' movement ecology, and shows how stable isotopes can be used to improve understanding of the relationships between baleen whales' foraging ecology and movements.

My one very minor quibble is with the first sentence of the Conclusions. The authors state, "Our work provides strong evidence that North Atlantic blue, fin and sei whales migrating through central Atlantic waters feed during winter or early spring in tropical and subtropical waters, and that this is a widespread strategy among migratory whales." I'd suggest rewording to "Our work provides strong evidence that North Atlantic blue, fin and sei whales migrating through

central Atlantic waters feed during winter or early spring in tropical and subtropical waters, and that this strategy is more prevalent among migratory whales than is currently accepted". Or something along those lines. My point being that we don't yet know that winter feeding is widespread amongst migratory whales, so that sentence isn't quite correct right now. I suspect that the authors are right, and the strategy is widespread, and I look forward to seeing work emulating their study, on whales in other ocean basins.

I have no real comments to make further, other than to congratulate the authors on a fine piece of work.

Reviewer: 2

Comments to the Author(s)

REVIEW OF

Stable isotopes reveal supplemental feeding and differences in winter habitats in blue, fin and sei whales migrating through the Azores,

This study investigate variability in isotope ratios from bulk skin tissue of blue, fin and sei whales collected near the Azores from 2002 to 2014. Authors also analysed faeces samples that were opportunistically collected during the same period of time. Prey data was obtained from published work, but data regarding the date of collection is not included anywhere in the manuscript or supplementary material. It is unknown if these prey data was collected within the same season (s) and years that whales skin samples were collected or from a different period of time. Authors used stable isotope analysis of C and N to understand some aspects of the feeding ecology of these whale species, but introduction about the application of this technique in cetaceans, and the meaning of $\delta^{13}\text{C}$ and $\delta^{15}\text{N}$ is poorly described in the introduction. There are various studies that had used stable isotope analysis in skin tissues from whales to understand variability in diet and habitat use by sex, feeding areas, maturity, etc. and identify migratory species from resident animals. There are also studies that included analysis of faeces and prey items. Introduce this information to the reader.

Author included only one paragraph describing isotope ratios, in particular, the spatial variability in primary producer values, but authors cited only one publication which refers to understand top predators movement (work of Graham et al. 2010). It seems that the authors are not reading important literature in regards to the factors driving variability in isotope ratios in plankton. At least, please cite the citations that Graham et al. 2010 included in their introduction to explain spatial variation in baseline values.

It is not clear what are the main goals of this work, e.g. if authors are trying to understand the contribution of prey items in the whales diet? Or if they are trying to understand the importance of the Azores as a feeding area for these species? Or if they are trying to identify which are the main feeding areas of migratory blue, sei and fin whales that are observed in the Azores? Or if the goal is to understand migration patterns? And whether these migratory patterns may contradict the classic migration routes for these species? Authors need to define clearly the objectives of their work and state their hypotheses.

In the methods, analysis, I also recommend to include Bayesian mixing models that are commonly used with stable isotope data. This analysis can help to ID the approximate feeding areas of these cetaceans (and discriminate others), rather than just saying that a mean value of X animal overlap with this other X mean value

Also, there are carbon and nitrogen isoscapes derived from zooplankton samples from the Atlantic, the work of McMahon et al. 2013, cited in this paper. Although these isoscapes also represent a snapshot within a specific period of time, authors can use this isoscape (in addition to the other data they included in the table S2) as a baseline to ID the main feeding areas of cetaceans and discriminate whether the waters near the Azores represent an important feeding area for some species or not. However, data from this work are not included as reference of potential prey of the current manuscript. I do not understand why would the authors exclude this work that could be very helpful to understand habitat use of whales? if there is a good reason, explain it briefly, it could be important.

Authors need to include collection date in Table 1, and annual sample size in the legend of figure 3.

Table S1: not integrated well in the analyses nor paper discussion, it seems to be just an isolated table no connected to the manuscript.

Table S2: include collection date or at least collection year and season

Table S3, why blue whales were not included in the analyses?

Time series- Another important issue is the poorly discussed temporal variability in skin samples, which were collected in different seasons and years. Author's results, and figure 3 (that I think they are very interesting) shows that there is a high to moderate temporal variation (and significant differences between years) for both C or N in fin or sei whales, but these pattern was not well discussed. Samples from blue whales are not shown, and I do recommend to include these data in the graphs (despite that they were not significantly different and sample size was small). I also recommend to explain this temporal variation at the very beginning of the discussion (rather than discuss these results at the end), please explain what does this isotopic variation through time mean for each species? Are whale switching feeding areas through time? And/OR Are whales preying on different items? AND/Or is there any changes in the baseline values? Why blue whales is the exception? Please discuss these results and provide insights before you show and discuss results from SEAc.

For SEAc analyses, you can include results showed in figure 2 and include it in figure 5 (instead of just providing mean and SD values of those whales). This would be a stronger way to present your data (and you can eliminate figure 2). More importantly, after showing and discussing results of temporal variability in whale isotope ratios, authors could discuss that despite the observed temporal variation in isotope ratios for Sei and fin whales, SEAc results suggest that whales have different isotope niches during those 12 years. Then, you can discuss if these results are associated to differences in habitat or prey, or both.

Figure 5, because there is no date of collection of prey items, it is really hard to understand and evaluate whether whales are feeding in prey from a given site and no from another site, because the zooplankton isotope ratios can vary largely through time. So, you need to show the variability in zooplankton isotope ratios collected in different years and season (if they were collected in different year and season) and could be included in the supplementary material, and then make a case that despite this temporal variation, the spatial variability in isotope ratios is still strong between feeding areas (as presumably shown in figure 5). Then, authors could use Bayesian mixing models to ID the main areas where whales were feeding.

Results, please include subheadings, and follow the same order to discuss results. Discussion section should not be an extension of results, please concise discussion. As it is, the discussion sections are very confusing, and hard to follow.

The very first sentence of the discussion, authors claimed to be the first one measuring isotope

rations in some specific areas, why this claim is the highlight of your discussion? This is not really interesting, it is more interesting to discuss better the main results of their study, and in relation to their hypotheses that were largely missing in this manuscript.

Page 6, L10, authors need to provide the expected baseline isotope values for upwelling areas vs nearshore areas, and oligotrophic areas in the Atlantic, and include labels in the figure 5 to ID which prey values represent oligotrophic, nearshore and upwelling areas. Otherwise, your discussion is very speculative.

General Evaluation: Because authors did not specify the goals of their work, it was really hard to follow their manuscript, and understand why they are using some analyses instead of other ones, and how one analyses connect with one specific goal. The reader has to guess most of the time.

Because there are many missing pieces of information in each section of the manuscript, the study reads weak and discussion read very speculative.

Author's Response to Decision Letter for (RSOS-181800.R0)

See Appendix A.

RSOS-181800.R1 (Revision)

Review form: Reviewer 1

Is the manuscript scientifically sound in its present form?

Yes

Are the interpretations and conclusions justified by the results?

Yes

Is the language acceptable?

Yes

Is it clear how to access all supporting data?

Yes

Do you have any ethical concerns with this paper?

No

Have you any concerns about statistical analyses in this paper?

No

Recommendation?

Accept as is

Comments to the Author(s)

My minor comments were dealt with in a satisfactory manner.

Review form: Reviewer 3

Is the manuscript scientifically sound in its present form?

Yes

Are the interpretations and conclusions justified by the results?

Yes

Is the language acceptable?

Yes

Is it clear how to access all supporting data?

Yes

Do you have any ethical concerns with this paper?

No

Have you any concerns about statistical analyses in this paper?

No

Recommendation?

Accept with minor revision (please list in comments)

Comments to the Author(s)

See attached file (Appendix B).

Decision letter (RSOS-181800.R1)

03-Jul-2019

Dear Dr Silva:

On behalf of the Editors, I am pleased to inform you that your Manuscript RSOS-181800.R1 entitled "Stable isotopes reveal winter feeding in different habitats in blue, fin and sei whales migrating through the Azores" has been accepted for publication in Royal Society Open Science subject to minor revision in accordance with the referee suggestions. Please find the referees' comments at the end of this email.

The reviewers and Subject Editor have recommended publication, but also suggest some minor revisions to your manuscript. Therefore, I invite you to respond to the comments and revise your manuscript.

- **Ethics statement**

- Data accessibility

If you wish to submit your supporting data or code to Dryad (<http://datadryad.org/>), or modify your current submission to dryad, please use the following link:
<http://datadryad.org/submit?journalID=RSOS&manu=RSOS-181800.R1>

- Competing interests

- Authors' contributions

- Acknowledgements

- Funding statement

Because the schedule for publication is very tight, it is a condition of publication that you submit the revised version of your manuscript before 12-Jul-2019. Please note that the revision deadline will expire at 00.00am on this date. If you do not think you will be able to meet this date please let me know immediately.

on behalf of Dr Asha de Vos (Associate Editor) and Kevin Padian (Subject Editor)
openscience@royalsociety.org

Associate Editor Comments to Author (Dr Asha de Vos):

Thank you for this great submission. We are happy to accept with minor revisions.

Reviewer comments to Author:
Reviewer: 1

Comments to the Author(s)
My minor comments were dealt with in a satisfactory manner.

Reviewer: 3

Comments to the Author(s)
See attached file

Author's Response to Decision Letter for (RSOS-181800.R1)

See Appendix C.

Decision letter (RSOS-181800.R2)

23-Jul-2019

Dear Dr Silva,

I am pleased to inform you that your manuscript entitled "Stable isotopes reveal winter feeding in different habitats in blue, fin and sei whales migrating through the Azores" is now accepted for publication in Royal Society Open Science.

on behalf of Dr Asha de Vos (Associate Editor) and Kevin Padian (Subject Editor)
openscience@royalsociety.org

Appendix A

Supplementary material from “Stable isotopes reveal winter feeding in different habitats in blue, fin and sei whales migrating through the Azores”

Methods

Potential range and seasonal movements of whale species

The summer feeding range of North Atlantic blue whales extends from the Scotian Shelf to Davis Strait in the west, and from Denmark Strait to Svalbard in the east [1]. Fin whales co-occur with blue whales in most of this range but on the eastern Atlantic their feeding grounds extend further south to the British islands, Bay of Biscay and Iberian coast [2]. Additionally, there is evidence that part of the fin whale population wintering in the western Mediterranean migrate towards the Atlantic during the summer months [3]. Satellite telemetry indicate that blue and fin whales seen in the Azores migrate to central Atlantic waters, between Greenland and Iceland [4]. A recent photo-identification match between the Azores and the Gulf of St. Lawrence indicates that blue whales seen in the Azores may spent the previous summer in either side of the North Atlantic basin [5]. All sei whales instrumented with satellite tags in the Azores migrated to the Labrador Sea, off Canada [6,7]. Still, two individuals were apparently heading east of Greenland when their signals were lost, suggesting individuals may use feeding grounds on both sides of the Atlantic [7].

The wintering grounds of these species remain unknown. Fin whales were acoustically detected along the Mid-Atlantic Ridge (16°-50° N, 24°-50°W) from late autumn through early spring, with higher detection rates north of 32°N during winter months [8]. Whaling records and historic and contemporary sightings of sei whales along the West African coast (from Morocco down to Senegal) [9,10] and of blue whales between Cape Verde and Mauritania in winter [10-12] suggest part of the population may winter off the

Northwest Africa coast.

Based on the above, we considered that blue and sei whales seen in the Azores may summer in the Northeast Atlantic (NEA) and Northwest Atlantic (NWA), and spend the 3-4 months preceding sampling in lower-latitude pelagic waters (Azores (AZ), North Africa,-NAF) or closer to the Northwest African coast (NAF-UPW). In the case of fin whales, we also considered the Iberia (IB), Bay of Biscay (BB) and western Mediterranean (MED) as plausible winter-spring habitats.

Table S1. Mean and standard deviation (SD) $\delta^{15}\text{N}$ and $\delta^{13}\text{C}$ values of potential prey taxa from the literature used to estimate the isotopic composition of prey groups in different regions, by season. Only studies that accounted for lipid through chemical extraction or mathematical correction and provided data for both $\delta^{15}\text{N}$ and $\delta^{13}\text{C}$ values, along with a descriptor of the variance and number of samples analysed, were included. Regions are: AZ-Azores, IB-Iberia, BB-Bay of Biscay, NAF-North Africa, NAF-UPW-North Africa Upwelling, NEA-Northeast Atlantic, NWA-Northwest Atlantic, MED-Mediterranean.

Region	Location	Prey group	Lowest taxonomic ID	Year	Season	$\delta^{15}\text{N}$		$\delta^{13}\text{C}$		N° samples	References
						Mean	SD	Mean	SD		
AZ	Azores	Copepods	Copepods	2009	Spring	5.16	0.63	-20.45	0.56	52	13
AZ	Azores	Euphausiids	Euphausiidae	2009	Spring	6.39	0.66	-19.94	0.66	27	13
BB	Bay of Biscay	Euphausiids	Meganyctiphanes norvegica	2001-10	Autumn	8.30	0.20	-19.80	0.20	15	14
IB	Iberia	Euphausiids	Meganyctiphanes norvegica	2001	Spring	6.35	0.56	-20.93	0.31	6	15
IB	Iberia	Euphausiids	Meganyctiphanes norvegica	2001	Autumn	7.55	0.57	-21.10	0.20	6	15
NAF	Cape Verde	Copepods	Undinula vulgaris	2012	Autumn	4.68	0.31	-19.49	0.38	3	16
NAF	Cape Verde	Euphausiids	Euphausiid	2012	Autumn	3.04	0.32	-18.60	0.16	3	16
NAF	Guinea Dome	Copepods	Undinula vulgaris	2012	Autumn	4.95	0.85	-19.14	0.13	6	16
NAF	Guinea Dome	Copepods	Calanoides sp.	2015	Spring	8.28	1.10	-19.37	0.80	13	17
NAF	Guinea Dome	Euphausiids	Euphausiacea	2015	Spring	7.79	1.06	-19.70	0.40	11	17
NAF	Guinea-Bissau - Liberia	Copepods	Undinula vulgaris	2012	Autumn	6.07	0.15	-19.97	0.06	3	18
NAF	Guinea-Bissau - Liberia	Euphausiids	Euphausiid	2012	Autumn	4.45	0.26	-19.90	0.24	4	18
NAF	Western Tropical Atlantic	Copepods	Undinula vulgaris	2012	Autumn	4.46	1.75	-19.47	0.90	23	16
NAF	Western Tropical Atlantic	Copepods	Calanoides sp.	2015	Spring	6.19	1.16	-19.50	0.60	19	17
NAF	Western Tropical Atlantic	Euphausiids	Euphausiid	2012	Autumn	4.23	1.60	-18.78	0.67	10	16
NAF	Western Tropical Atlantic	Euphausiids	Euphausiacea	2015	Spring	6.60	0.96	-19.22	0.73	24	17
NAF	Western Tropical Atlantic	Euphausiids	Meganyctiphanes norvegica	2015	Spring	8.00	0.81	-19.17	0.05	2	17
NAF	Western Tropical Atlantic	Euphausiids	Thysanoessa sp.	2015	Spring	8.15		-20.54		1	17
NAF-UPW	Saharan bank	Copepods	Calanoides sp.	2015	Spring	5.57	1.15	-17.93	1.35	7	17
NAF-UPW	Saharan bank	Euphausiids	Euphausiid	2012	Autumn	5.98	0.31	-21.30	0.20	4	18
NAF-UPW	Saharan bank	Euphausiids	Euphausiacea	2015	Spring	6.25	1.58	-18.16	1.53	14	17
NAF-UPW	Senegal-Mauritania	Copepods	Calanoides carinatus	2012	Autumn	6.80	0.91	-17.70	0.45	8	18
NAF-UPW	Senegal-Mauritania	Copepods	Undinula vulgaris	2012	Autumn	4.62	0.52	-19.06	0.41	19	16, 18

NAF-UPW	Senegal-Mauritania	Copepods	Calanoides sp.	2015	Spring	8.30	1.09	-18.34	1.18	27	17
NAF-UPW	Senegal-Mauritania	Euphausiids	Euphausiid	2012	Autumn	6.14	1.59	-19.16	0.49	10	16, 18
NAF-UPW	Senegal-Mauritania	Euphausiids	Euphausiacea	2015	Spring	8.63	1.16	-18.51	1.71	19	17
NEA	Iceland Sea	Euphausiids	Meganyctiphanes norvegica	2007	Summer	7.50	0.52	-20.40	0.35	3	19
NEA	Iceland Sea	Euphausiids	Thysanoessa inermis	2007	Summer	8.70	0.17	-20.20	0.69	3	19
NEA	Iceland Sea	Euphausiids	Thysanoessa longicaudata	2007	Summer	9.00	0.17	-22.10	0.17	3	19
NEA	Spitzbergen	Copepods	Calanus finmarchicus	2007	Summer	8.60	0.20	-21.60	0.01	4	20
NEA	Spitzbergen	Copepods	Calanus hyperboreus	2007	Summer	6.70	1.00	-22.20	0.26	4	21
NEA	Spitzbergen	Euphausiids	Euphausiidae	2007	Summer	8.30	0.10	-20.00	0.11	4	21
NEA	Svalbard	Copepods	Calanus hyperboreus	2003	Summer	8.83	0.85	-23.76	1.37	104	22
NEA	Svalbard	Copepods	Calanus glacialis	2003	Summer	8.40	1.31	-22.58	1.46	146	22
NEA	Svalbard	Copepods	Calanus glacialis	2008	Summer	9.10	0.25	-20.83	0.27	9	20
NEA	Svalbard	Copepods	Calanus finmarchicus	2008	Summer	7.90	0.10	-21.20	0.10	3	20
NEA	Svalbard	Copepods	Calanus hyperboreus	2008	Summer	8.44	0.42	-20.66	0.27	7	20
NEA	Svalbard	Euphausiids	Meganyctiphanes norvegica	2003	Summer	11.50	0.35	-21.70	0.17	3	22
NEA	Svalbard	Euphausiids	Thysanoessa longicaudata	2003	Summer	10.30	0.51	-21.70	0.51	26	22
NEA	Svalbard	Euphausiids	Thysanoessa inermis	2003	Summer	9.40	1.20	-22.30	1.20	16	22
NEA	Svalbard	Euphausiids	Euphausiidae	2008	Summer	8.85	0.33	-20.80	0.57	6	20
NWA	Gulf St Lawrence	Copepods	Calanus finmarchicus	2006	Summer	8.30	0.32	-22.70	0.32	10	23
NWA	Gulf St Lawrence	Copepods	Calanus hyperboreus	2006	Summer	8.00	0.22	-22.40	0.22	5	23
NWA	Gulf St Lawrence	Euphausiids	Meganyctiphanes norvegica	2006	Summer	9.20	0.49	-21.10	0.24	6	23
NWA	SW Greenland	Euphausiids	Meganyctiphanes norvegica	2003	Summer	8.50	0.40	-19.00	0.00	2	24
NWA	Western Greenland	Copepods	Calanus spp.	2010	Summer	8.39	1.27	-20.97	0.84	38	25
NWA	Western Greenland	Euphausiids	Meganyctiphanes norvegica	2010	Summer	11.37	1.23	-21.96	0.45	40	25
NWA	Western Greenland	Euphausiids	Thysanoessa inermis	2010	Summer	10.04	0.95	-21.71	0.67	59	26
NWA	Western Greenland	Euphausiids	Thysanoessa longicaudata	2010	Summer	8.64	1.30	-22.60	0.67	21	26
NWA	Western Greenland	Euphausiids	Thysanoessa raschii	2010	Summer	9.26	0.66	-21.49	0.54	61	26
MED	Balearic Sea	Euphausiids	Meganyctiphanes norvegica	2009	Winter	6.91	0.74	-19.68	0.33	5	27

Table S2. Estimates of the mean and standard deviation (SD) $\delta^{15}\text{N}$ and $\delta^{13}\text{C}$ values of regions used in Bayesian stable isotope mixing models run for blue, fin and sei whales. The model for sei whales included isotopic data from copepods belonging to family Calanidae; the models for blue and fin whales included isotopic data from euphausiids belonging to family Euphausiidae. Regions are: AZ-Azores, IB-Iberia, BB-Bay of Biscay, NAF-North Africa, NAF-UPW-North Africa Upwelling, NEA-Northeast Atlantic, NWA-Northwest Atlantic, MED-Mediterranean.

Regions	Season	Years	$\delta^{15}\text{N}$		$\delta^{13}\text{C}$		N
			Mean	SD	Mean	SD	
Blue whale model							
NAF-UPW	Autumn	2012	6.09	1.33	-19.77	1.08	14
NAF-UPW	Spring	2015	7.73	1.69	-18.76	1.02	33
NAF	Autumn	2012	4.07	1.31	-19.01	0.73	17
NAF	Spring	2015	7.06	1.12	-19.39	0.68	38
AZ	Spring	2009	6.39	0.66	-19.94	0.66	27
NEA	Summer	2003, 2007, 2008	9.60	0.79	-21.55	0.72	64
NWA	Summer	2003, 2006, 2010	9.78	0.96	-21.79	0.36	189
Fin whale model							
NAF-UPW	Autumn	2012	6.09	1.33	-19.77	1.08	14
NAF-UPW	Spring	2015	7.73	1.69	-18.76	1.02	33
NAF	Autumn	2012	4.07	1.31	-19.01	0.73	17
NAF	Spring	2015	7.06	1.12	-19.39	0.68	38
AZ	Spring	2009	6.39	0.66	-19.94	0.66	27
NEA	Summer	2003, 2007, 2008	9.60	0.79	-21.55	0.72	64
NWA	Summer	2003, 2006, 2010	9.78	0.96	-21.79	0.36	189
MED	Winter	2009	6.91	0.74	-19.68	0.33	5
IB	Spring	2001	6.35	0.56	-20.93	0.31	
IB	Autumn	2001	7.55	0.57	-21.10	0.20	
BB	Autumn	2001-2010	8.3	0.20	-19.80	0.20	
Sei whale model							
NAF-UPW	Autumn	2012	5.26	1.20	-18.66	0.76	27
NAF-UPW	Spring	2015	7.71	1.56	-18.35	1.09	34
NAF	Autumn	2012	4.70	1.52	-19.46	0.76	40
NAF	Spring	2015	7.04	1.53	-19.45	0.68	32
AZ	Spring	2009	5.16	0.63	-20.45	0.56	52
NEA	Summer	2003, 2007, 2008, 2013	8.52	1.10	-22.89	1.34	277
NWA	Summer	2006, 2010	8.36	1.07	-22.04	0.85	91

12 References

- 13 1 Sears R, Perrin WF. 2018 Blue Whale: *Balaenoptera musculus*. In *Encyclopedia of Marine*
14 *Mammals* 3rd edition (eds. B Würsig, KM Kovacs, JGM Thewissen). *Academic Press*
15 (doi:10.1016/C2015-0-00820-6)
- 16 2 Edwards EF, Hall C, Moore TJ, Sheredy C, Redfern JV. 2015 Global distribution of fin
17 whales *Balaenoptera physalus* in the post-whaling era (1980–2012). *Mammal Rev.* **45**,
18 197–214. (doi:10.1111/mam.12048)
- 19 3 Geijer CKA, Notarbartolo di Sciara G, Panigada S. 2016 Mysticete migration revisited: Are
20 Mediterranean fin whales an anomaly? *Mammal Rev.* **46**, 284–296.
21 (doi:10.1111/mam.12069)
- 22 4 Silva MA, Prieto R, Jonsen I, Baumgartner MF, Santos RS. 2013 North Atlantic blue and fin
23 whales suspend their spring migration to forage in middle latitudes: Building up energy
24 reserves for the journey? *PLoS One.* **8**, e76507. (doi:10.1371/journal.pone.0076507)
- 25 5 Sears R, Vikingsson G, Santos R, Steiner L, Silva M, Ramp C. 2015 Comparison of northwest
26 Atlantic (NWA) and northeast Atlantic (NEA) blue whale (*Balaenoptera musculus*)
27 photo-identification catalogues. 21st Biennial Conference on the Biology of Marine
28 Mammals. San Francisco CA.
- 29 6 Olsen E, Budgell P, Head E, Kleivane L, Nottestad L, Prieto R, Silva MA, Skov H,
30 Vikingsson GA, Waring GT et al. 2009 First satellite-tracked long-distance movement of
31 a sei whale (*Balaenoptera borealis*) in the North Atlantic. *Aquat Mamm.* **35**, 313–318.
32 (doi:10.1578/AM.35.3.2009.31)
- 33 7 Prieto R, Silva MA, Waring GT, Gonçalves JMA. 2014 Sei whale movements and behaviour
34 in the north Atlantic inferred from satellite telemetry. *Endang Species Res.* **26**, 103–113.
35 (doi:10.3354/esr00630)
- 36 8 Nieuwkirk SL, Mellinger DK, Moore SE, Klinck K, Dziak RP, Goslin J. 2012 Sounds from
37 airguns and fin whales recorded in the mid-Atlantic Ocean, 1999–2009. *J Acoust Soc Am.*
38 **131**, 1102–1112. (doi:10.1121/1.3672648)
- 39 9 Prieto R, Janiger D, Silva MA, Waring GT, Gonçalves JM. 2012 The forgotten whale: a
40 bibliometric analysis and literature review of the North Atlantic sei whale *Balaenoptera*
41 *borealis*. *Mammal Rev.* **42**: 235–272. (doi:10.1111/j.1365-2907.2011.00195.x)
- 42 10 Baines M, Reichelt M. 2014 Upwellings, canyons and whales: An important winter habitat
43 for balaenopterid whales off Mauritania, northwest Africa. *J Cetacean Res Manag.* **14**,
44 57–67.
- 45 11 Jonsgård A. 1966 The distribution of Balaenopteridae in the North Atlantic ocean. In
46 *Whales, dolphins and porpoises.* (ed. KS Norris). Los Angeles, California: University of
47 California Press.
- 48 12 Reeves RR, Smith TD, Josephson EA, Clapham PJ, Woolmer G. 2004 Historical
49 observations of humpback and blue whales in the North Atlantic ocean: clues to
50 migratory routes and possibly additional feeding grounds. *Mar Mamm Sci.* **20**, 774–786.
51 (doi:10.1111/j.1748-7692.2004.tb01192.x)
- 52 13 Colaço A, Giacomello E, Porteiro F, Menezes GM. 2013 Trophodynamic studies on the
53 Condor seamount (Azores, Portugal, North Atlantic). *Deep Sea Res II.* **98**, 178–189.
54 (doi:10.1016/j.dsr2.2013.01.010)
- 55 14 Chouvelon T, Spitz J, Caurant F, Mèndez-Fernandez P, Chappuis A, Laugier F, Le Goff E,
56 Bustamante P. 2012 Revisiting the use of $\delta^{15}\text{N}$ in meso-scale studies of marine food webs
57 by considering spatio-temporal variations in stable isotopic signatures – The case of an
58 open ecosystem: The Bay of Biscay (North-East Atlantic). *Prog Oceanogr.* **101**, 92–105.
59 (doi:10.1016/j.pocean.2012.01.004)

- 60 15 Bentaleb I, Martin C, Vrac M, Mate B, Mayzaud P, Siret D, de Stephanis R, Guinet C. 2011
61 Foraging ecology of Mediterranean fin whales in a changing environment elucidated by
62 satellite tracking and baleen plate stable isotopes. *Mar Ecol Prog Ser.* **438**, 285-302. (doi:
63 10.3354/meps09269)
- 64 16 Sandel V, Kiko R, Brandt P, Dengler M, Stemmann L, Vandromme P, Sommer U, Hauss
65 H. 2015 Nitrogen fuelling of the pelagic food web of the Tropical Atlantic. *PLoS ONE.*
66 **10**, e0131258. (doi:10.1371/journal.pone.0131258)
- 67 17 Bode A, Hernandez-Leon S. 2018. Trophic diversity of plankton in epipelagic and
68 mesopelagic layers of the tropical and equatorial Atlantic determined with stable
69 isotopes. *Diversity.* **10**, 48. (doi:10.3390/d10020152)
- 70 18 Bode M, Hagen W, Schukat A, Teuber L, Fonseca-Batista D, Dehairs F, Auel H. 2015
71 Feeding strategies of tropical and subtropical calanoid copepods throughout the eastern
72 Atlantic Ocean – latitudinal and bathymetric aspects. *Prog Oceanogr.* **138**, 268-282.
73 (doi:10.1016/j.pocean.2015.10.002)
- 74 19 Petursdottir H, Gislason A. 2009 Trophic interactions and energy flow within the pelagic
75 ecosystem in the Iceland Sea. *ICES CM* 2009/A:08
- 76 20 Hallanger IG, Ruus A, Warner NA, Herzke D, Evenset A, Schøyen M, Gabrielsen GW,
77 Borgå K. 2011. Differences between Arctic and Atlantic fjord systems on
78 bioaccumulation of persistent organic pollutants in zooplankton from Svalbard. *Sci Total*
79 *Environ.* **15**, 2783-2795. (doi: 10.1016/j.scitotenv.2011.03.015)
- 80 21 Hallanger IG, Warner NA, Ruus A, Evenset A, Christensen G, Herzke D, Gabrielsen GW,
81 Borgå K. 2011. Seasonality in contaminant accumulation in Arctic marine pelagic food
82 webs using trophic magnification factor as a measure of bioaccumulation. *Environ*
83 *Toxicol Chem.* **30**, 1026-1035. (doi: 10.1002/etc.488)
- 84 22 Sørdeide JE, Hop H, Carroll ML, Falk-Petersen S, Hegseth EN. 2013 Seasonal food web
85 structures and sympagic–pelagic coupling in the European Arctic revealed by stable
86 isotopes and a two-source food web model. *Prog Oceanogr.* **71**, 59-87. (doi:
87 10.1016/j.pocean.2006.06.001)
- 88 23 Lavoie RA, Hebert CE, Rail JF, Braune BM, Yumvihoze E, Hill LG, Lean DRS. 2010
89 Trophic structure and mercury distribution in a Gulf of St. Lawrence (Canada) food web
90 using stable isotope analysis. *Sci Total Environ.* **408**, 5529-5539. (doi:
91 10.1016/j.scitotenv.2010.07.053)
- 92 24 Møller P. 2006 Lipids and stable isotopes in marine food webs in West Greenland. Trophic
93 relations and health implications. Phd thesis. National Environmental Research Institute,
94 Denmark.
95 (http://www2.dmu.dk/1_viden/2_Publikationer/3_Ovrige/rapporter/phd_PEM.pdf)
- 96 25 Bode A, Agersted MD, Nielsen TG, Basedow SL, Petursdottir H, Gislason A. 2014. Stable
97 C and N isotopes and percent C as well as N composition of plankton from the North
98 Atlantic. PANGAEA. (doi: 10.1594/PANGAEA.837299)
- 99 26 Valls MM. 2017 Trophic ecology in marine ecosystems from the Balearic Sea (Western
100 Mediterranean). PhD thesis. Universitat de les Illes Balears.
101 (<https://www.tesisenred.net/handle/10803/461496>)
- 102 27 Barría C, Navarro J, Coll M. 2017. Trophic habits of an abundant shark in the northwestern
103 Mediterranean Sea using an isotopic non-lethal approach. *Estuar Coast Shelf Sci.* **207**,
104 383-390. (doi:1016/j.ecss.2017.08.021)
- 105

Results

Table S3 Summary of results of one-way ANOVAs and Student's *t*-test assessing differences in $\delta^{15}\text{N}$ and $\delta^{13}\text{C}$ values among whale species (blue, fin and sei whales) and among sexes, seasons and years for fin and sei whales. Only factors for which the PERMANOVA indicated a statistically significant effect on stable isotopes were tested with univariate analysis (except for the comparison between spring and summer isotope values for fin whales). Statistically significant ($p < 0.05$) pairwise comparisons based on post hoc Tukey HSD test are shown in the table.

Analyses	Factor	df	$\delta^{15}\text{N}$ value			$\delta^{13}\text{C}$ value		
			Student's t or ANOVA F	P	Post hoc Tukey HSD	Student's t or ANOVA F	P	Post hoc Tukey HSD
Inter-species		2, 92	F =6.21	0.003	fin–sei	F =44.61	<0.001	blue–fin, blue–sei, fin–sei
Fin whale	Sex	40	t =0.37	0.713		t =1.60	0.118	
	Season	38	t =-0.92	0.365		t =0.81	0.423	
	Year	3, 34	F =5.89	0.002	2008–2010, 2008–2013, 2008–2014	F =27.25	<0.001	2008–2013, 2008–2014, 2010–2014
Sei whale	Season	2, 27	F =0.32	0.731		F =28.86	<0.001	spring–summer, spring–autumn,
	Year	5, 27	F =9.74	<0.001	2005–2014, 2008–2014, 2009–2014	F =4.49	0.004	2008–2012, 2008–2014, 2009–2012

Copepods

Euphausiids

Figure S1 - Mean and standard deviation (SD) $\delta^{15}\text{N}$ and $\delta^{13}\text{C}$ values of potential prey items from different regions and seasons. Copepods are shown in the top plot and euphausiids in the bottom plot. Regions are represented by different colours: AZ-Azores (yellow), IB-Iberia

(pink), BB-Bay of Biscay (light blue), NAF-North Africa (orange), NAF-UPW-North Africa Upwelling (red), NEA-Northeast Atlantic (dark blue), NWA-Northwest Atlantic (green), MED-Mediterranean (gray). Seasons are represented by different symbols: Winter (diamond), Spring (triangle), Summer (circle), Autumn (square). Smaller symbols represent mean (SD) isotopic values of individual taxa and larger symbols represent the weighted average (SD) by region and season.

Table S4 – Percentage mean and SD contribution of different regional sources to the diet of blue, fin and sei whales estimated using Bayesian isotopic mixing models. Regions are: AZ-Azores, IB-Iberia, BB-Bay of Biscay, NAF-North Africa, NAF-UPW-North Africa Upwelling, NEA-Northeast Atlantic, NWA-Northwest Atlantic, MED-Mediterranean.

Region-season	Season	Blue whale		Fin whale		Sei whale			
		Mean	SD	Mean	SD	Spring		Summer	
		Mean	SD	Mean	SD	Mean	SD	Mean	SD
NAF-UPW-aut	Autumn	20.0	16.7	4.5	4.5	39.3	15.9	27.8	14.7
NAF-UPW-spr	Spring	7.5	5.6	2.7	2.1	18.8	9.5	19.8	10.9
NAF-aut	Autumn	27.9	12.8	4.1	3.6	13.5	10.7	13.1	8.9
NAF-spr	Spring	10.3	8.5	3.2	2.7	11.7	9.1	12.6	8.8
AZ	Spring	18.1	16.6	4.3	4.2	7.8	5.8	14.0	11.0
NEA	Summer	8.3	5.8	3.1	2.3	4.2	2.6	5.9	4.3
NWA	Summer	8.0	5.7	3.1	2.4	4.8	3.1	6.8	4.9
MED	Winter			3.6	3.2				
IB-spr	Spring			63.6	11.5				
IB-aut	Autumn			4.8	5.3				
BB-aut	Autumn			3.0	2.3				

Appendix B

Manuscript RSOS-181800.R1
by Silva et al.
Royal Society for Open Science

Stable isotopes reveal winter feeding in different habitats in blue, fin and sei whales migrating through the Azores

This study examines winter feeding in three species of baleen whales by comparing carbon and nitrogen isotope ratios in skin biopsy samples, which provide information over the past 2-3 mo of feeding behaviour.

This manuscript is generally very well written. The statistical analyses appear adequate, conclusions are nuanced, literature cited is highly relevant, and caveats clearly identified. I consider my comments generally minor, except for one point that require specific attention:

- 1) Lipids were extracted from samples using strong solvents to avoid biases of $\delta^{13}\text{C}$. Given the mixed results obtained in other studies about the effect of lipid-extraction on $\delta^{15}\text{N}$ values, there is a need to provide a justification for using lipid-extracted samples for $\delta^{15}\text{N}$ analysis instead of untreated samples.

Specific (minor) comments:

- There is a total mismatch between figure captions and contents.
- The paper says that samples were collected between 2002 and 2014 (first line in Methods), but the figures (fig 2) present results only from 2005.
- p. 2, line 26: do you mean 'that remain on the animal for the entire annual migratory cycle'.
- p. 2, line 46: should the term 'type' be changed to 'pathways'?
- p. 2, line 49: $\delta^{13}\text{C}$ decline from northern to southern latitudes yes, but within each hemisphere (so it is lower near the Equator and higher both toward the north and south poles).
- p.4, line 28: from Morocco to Senegal, not the Morocco... And please add a reference to fig. 1 here.
- p.4, line 37: 'of whale sampling' not of 'whale samples'
- p.4, line 44: delete 'were'
- p. 5, line 50: isotope or isotopic space?
- p.6, line 2: considerable not considerably. And should this be 'within-region' or 'within-regional'?
- p.6, line 16: reword: '...mainly used food resources available in NAF and NAF-UPW during autumn, and in the Azores during spring'.
- p. 6, Line 32: delete either 'the' or 'these'.
- p.6, line 43: this sentence implies there you expected a sex difference in blue whales. Why? Another sentence needs to be added to explain the expected difference in habitat use between males and breeding females during winter.
- p.7, line 3: why is the Suess effect likely to be negligible? You have samples spanning over more than a decade. What is the decadal change in $\delta^{13}\text{C}$ expected from Suess. Please add a sentence to support your statement.
- p.7 line 50: 'feeding' not 'feedings' habits

- p.7 lines 54-56: not obvious from the figure S1 that copepods are at lower TP than krill. Also, krill contains several species; might be true for *Meganyctiphanes norvegica*, but *Thysanoessa* species might actually be isotopically similar to copepods. Need to beef up this argument if it is to be kept here.
- p.8, line 26: 'compared to values from the Gulf of St. Lawrence'
- Figure 3 (with caption being fig.4) – julian day vs dC and DN in sei whale: the caption say top-bottom, but they are left-right panels
- Supplementary material: a few typos in the text: first paragraph: may spend (not spent) the previous summer on either side (not in either side).
- Table S2: some sample sizes missing

Appendix C

Response to Referees

Reviewer #1 was happy with our previous revision.

Response to Reviewer #3

Reviewer#3: I consider my comments generally minor, except for one point that require specific attention:

1) Lipids were extracted from samples using strong solvents to avoid biases of $\delta^{13}\text{C}$. Given the mixed results obtained in other studies about the effect of lipid-extraction on $\delta^{15}\text{N}$ values, there is a need to provide a justification for using lipid-extracted samples for $\delta^{15}\text{N}$ analysis instead of untreated samples.

R: *We agree with the Reviewer that the issue of potential effects of lipid extraction on $\delta^{15}\text{N}$ is far from being settled and included a few sentences in the Material and Methods section (Page 3, Lines 33-43) introducing the problem and explaining why we had to analyse $\delta^{15}\text{N}$ from lipid-extracted samples:*

“Lipids are depleted in ^{13}C and typically have $\delta^{13}\text{C}$ values that are more negative than those of proteins and carbohydrates within an organism [29], which can introduce considerable bias in SIAs. A common approach to deal with the potential lipid effect on $\delta^{13}\text{C}$ values, is to remove lipids from tissues by chemical lipid extraction [32]. However, lipid extraction may also alter $\delta^{15}\text{N}$ values of tissues, although the magnitude and direction of change is species and tissue dependent. In the case of baleen whale skin, some studies reported a significant decline in $\delta^{15}\text{N}$ values after lipid extraction [43], while other studies documented a significant increase or no change in $\delta^{15}\text{N}$ values for the same whale species [54]. Similarly to what was observed in the muscle tissue of terrestrial mammals [55], changes in $\delta^{15}\text{N}$ in baleen whale skin due to lipid extraction were generally small and within the analytical precision typical of $\delta^{15}\text{N}$ measurements [43, 54]. Nonetheless, a way to avoid potential effects on $\delta^{15}\text{N}$ is to measure $\delta^{13}\text{C}$ in lipid-extracted tissue and $\delta^{15}\text{N}$ in non-extracted tissue [54]. Obviously, this requires that a larger amount of sample is available to conduct separate analyses. Unfortunately, this was not the case in the present study, where each skin sample was already split for SIA and genetic analyses.”

Reviewer#3: Specific minor comments - the Reviewer noted several grammar errors throughout the ms and made specific suggestions to correct them.

R: *We adopted all the recommended changes and we thank the reviewer for taking his/her time to revise the English of the ms, including of the Supplementary material.*

Reviewer#3: The paper says that samples were collected between 2002 and 2014 (first line in Methods), but the figures (fig 2) present results only from 2005.

R: *We are thankful to the reviewer for pointing out this discrepancy. Figure 2 should show only the years for which there were at least 2 skin samples analysed, which were the years used in the inter-annual statistical analyses. However, when we added the plot for the blue whales as recommended by Reviewer#2, we inadvertently included the year 2010, which had only one sample.*

Species plots in figure 2 were corrected to only include years with $n \geq 2$ and we added a sentence in the caption of the figure explaining this option.

Reviewer#3: p.6, line 43: this sentence implies there you expected a sex difference in blue

whales. Why? Another sentence needs to be added to explain the expected difference in habitat use between males and breeding females during winter.

R: We wouldn't expect sex differences in habitat use but we agree with the reviewer that the way the sentence was written gave that impression. We reworded this sentence and the one in Pg 7, line 37 to separate the discussion of temporal and sex differences in isotope values.

Reviewer#3: p.7, line 3: why is the Suess effect likely to be negligible? You have samples spanning over more than a decade. What is the decadal change in $\delta^{13}\text{C}$ expected from Suess. Please add a sentence to support your statement.

R: Fin whale samples used in statistical analyses spanned 6 years and the decline in $\delta^{13}\text{C}$ over these 6 years was $\sim 2\%$, 80 times the highest estimates of the annual rate of change attributed to the Suess effect. Following the reviewer recommendation, we added a sentence in Pg 7, lines 6-10 supporting our argument:

“Similarly, the estimated rate of change in $\delta^{13}\text{C}$ of dissolved inorganic carbon (DIC) in the North Atlantic attributable to the Suess effect (the decrease in the $\delta^{13}\text{C}$ values of atmospheric CO_2 from the combustion of fossil fuels) ranges between -0.007% per year in polar regions and -0.025% per year in subtropical regions [69-71]. This value is too low to account for the annual differences in fin whale $\delta^{13}\text{C}$ values.”

Reviewer#3: p.7 lines 54-56: not obvious from the figure S1 that copepods are at lower TP than krill. Also, krill contains several species; might be true for *Meganyctiphanes norvegica*, but *Thysanoessa* species might actually be isotopically similar to copepods. Need to beef up this argument if it is to be kept here.

R: We agree that there are species-specific, spatial and temporal differences in $\delta^{15}\text{N}$ within copepods and krill. On average, however, copepods tend to have lower $\delta^{15}\text{N}$ than krill (Hansen et al., 2012; Agersted et al., 2014) and estimates of regional prey isotopic values compiled from the literature agree with this (Table S2). Following the reviewer recommendation, we reworded the sentence to put it more explicitly:

“A comparison of isotopic values of calanoid copepods and euphausiids compiled from the literature shows that, with a single exception (NAF in autumn), mean $\delta^{15}\text{N}$ values of copepods were consistently lower (by 0.2-1.42‰) than those of euphausiids sampled in the same region and season. Thus, the higher reliance of sei whales on copepods would explain their lower $\delta^{15}\text{N}$ values relative to fin and blue whales. “

*Hansen JH, Hedeholm RB, Süksen K, Christensen JT, Grønkvær P. 2012. Spatial variability of carbon ($\delta^{13}\text{C}$) and nitrogen ($\delta^{15}\text{N}$) stable isotope ratios in an Arctic marine food web. *Mar Ecol Prog Ser* 467: 47–59.*

*Agersted MD, Bode A, Nielsen TG. 2014 Trophic position of coexisting krill species: a stable isotope approach. *Mar Ecol Prog Ser* 516: 139–151.*